# Likelihood Ratio Confidence Sets for Sequential Decision Making

**Nicolas Emmenegger**[*]
ETH Zürich

**Mojmír Mutný**[*]
ETH Zürich

**Andreas Krause**
ETH Zürich

## Abstract

Certifiable, adaptive uncertainty estimates for unknown quantities are an essential ingredient of sequential decision-making algorithms. Standard approaches rely on problem-dependent concentration results and are limited to a specific combination of parameterization, noise family, and estimator. In this paper, we revisit the likelihood-based inference principle and propose to use *likelihood ratios* to construct *any-time valid* confidence sequences without requiring specialized treatment in each application scenario. Our method is especially suitable for problems with well-specified likelihoods, and the resulting sets always maintain the prescribed coverage in a model-agnostic manner. The size of the sets depends on a choice of estimator sequence in the likelihood ratio. We discuss how to provably choose the best sequence of estimators and shed light on connections to online convex optimization with algorithms such as Follow-the-Regularized-Leader. To counteract the initially large bias of the estimators, we propose a reweighting scheme that also opens up deployment in non-parametric settings such as RKHS function classes. We provide a *non-asymptotic* analysis of the likelihood ratio confidence sets size for generalized linear models, using insights from convex duality and online learning. We showcase the practical strength of our method on generalized linear bandit problems, survival analysis, and bandits with various additive noise distributions.

## 1 Introduction

One of the main issues addressed by machine learning and statistics is the estimation of an unknown *model* from noisy observations. For example, in supervised learning, this might concern learning the dependence between an input (covariate) $x$ and a random variable (observation) $y$. In many cases, we are not only interested in an estimate $\hat{\theta}$ of the true model parameter $\theta_\star$, but instead in a set of plausible values that $\theta_\star$ could take. Such confidence sets are of tremendous importance in sequential decision-making tasks, where uncertainty is used to drive exploration or risk-aversion needs to be implemented, and covariates are iteratively chosen based on previous observations. This setting includes problems such as bandit optimization, reinforcement learning, or active learning. In the former two, the confidence sets are often used to solve the *exploration-exploitation* dilemma and more generally influence the selection rule (Mukherjee et al., 2022), termination rule (Katz-Samuels and Jamieson, 2020), exploration (Auer, 2002) and/or risk-aversion (Makarova et al., 2021).

When we interact with the environment by gathering data sequentially based on previous confidence sets, we introduce correlations between past noisy observations and future covariates. Data collected in this manner is referred to as *adaptively gathered* (Wasserman et al., 2020). Constructing estimators, confidence sets, and hypothesis tests for such non-i.i.d. data comes with added difficulty. Accordingly, and also for its importance in light of the reproducibility crisis (Baker, 2016), the task has attracted significant attention in the statistics community in recent years (Ramdas et al., 2022).

---

[*]Equal contribution.

37th Conference on Neural Information Processing Systems (NeurIPS 2023).

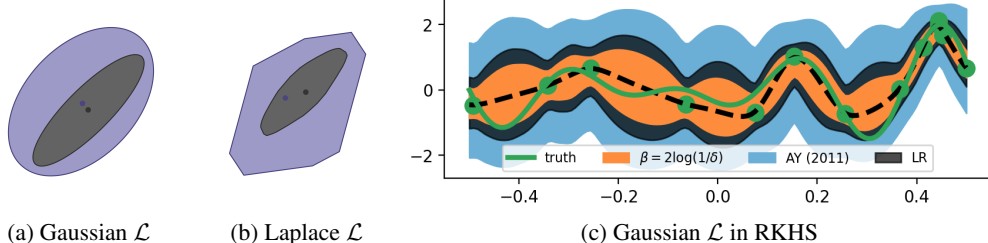

| (a) Gaussian $\mathcal{L}$ | (b) Laplace $\mathcal{L}$ | (c) Gaussian $\mathcal{L}$ in RKHS |

Figure 1: (a) and (b) show examples of confidence sets defined via level sets of the log-likelihood function in 2D at two dataset sizes, for Gaussian (a) and Laplace (b) likelihoods respectively. The sets inherit the geometry of the likelihood, and are not always ellipsoidal. (c) shows confidence bands on an RKHS function in a bandit game searching for the optimum. We compare prior work on confidence sets (Abbasi-Yadkori et al., 2011), our LR sets, and a common heuristic (orange). Our sets are nearly as small as the commonly used heuristic, but have provable coverage and can vastly improve sequential decision-making tasks such as bandits by quickly eliminating hypotheses.

Instead of deriving explicit concentration inequalities around an online estimator, we construct confidence sets *implicitly* defined by an inclusion criterion that is easy to evaluate in a computationally efficient manner and requires little statistical knowledge to implement. Roughly speaking, given a model $p_\theta(y \mid x)$ that describes the conditional dependence of the observation $y$ given the covariate $x$ under parameter $\theta$, we will build sets based on a *weighted* modification of the sequential likelihood ratio statistic (Robbins et al., 1972; Wasserman et al., 2020)

$$R_t(\theta) := \frac{\mathcal{L}_t(\{\hat{\theta}_s\}_{s=1}^t)}{\mathcal{L}_t(\theta)} := \frac{\prod_{s=1}^t p_{\hat{\theta}_s}^{w_s}(y_s \mid x_s)}{\prod_{s=1}^t p_\theta^{w_s}(y_s \mid x_s)}, \tag{1}$$

where $\{\hat{\theta}_s\}_s$ is a running estimator sequence that we are free to choose, but which may only depend on *previously* collected data. Parameters $\theta$ for which this statistic is small, i.e., for which $R_t(\theta) \leq 1/\alpha$ will be included in the set (and considered *plausible*). Examples of sets in a parametric and non-parametric setting are shown in Figure 1. The weighting terms $w_s \in (0, 1]$ are crucial for dealing with inherent irregularities of many conditional observation models but can be flexibly chosen. Classically, these are set to $w_s = 1$. The full exposition of our method with choice of estimators and weights is given in Section 2. Apart from being easy to use and implement, our approach also comes with performance guarantees. These sets maintain a provable $1 - \alpha$ coverage – a fact we establish using Ville's inequality for supermartingales (Ville, 1939), which is known to be essentially tight for martingales (see Howard et al., 2018, for a discussion). Therefore, in stark contrast to alternate methods, our confidence sequence is *fully data-dependent*, making it empirically tighter than competing approaches. Despite the rich history of sequential testing and related confidence sets going back to Wald (1945) and Robbins et al. (1972), these sets have found little use in the interactive machine learning community, which is a gap we fill in the present paper.

**Contributions** In this work, we revisit the idea of using likelihood ratios to generate anytime-valid confidence sets. The main insight is that whenever the likelihood of the noise process is known, the likelihood ratio confidence sets are fully specified. They inherit their geometry from the likelihood function, and their size depends on the quality of our estimator sequence. We critically evaluate the likelihood ratio confidence sets and, in particular, we shed light on the following aspects: **Firstly**, for generalized linear models, we *theoretically* analyze the geometry of the LR confidence sets under mild assumptions. We show their geometry is dictated by Bregman divergences of exponential families (Chowdhury et al., 2022). **Secondly**, we show that the size of the confidence set is dictated by an online prediction game. The size of these sets depends on a sequence of estimators $\{\hat{\theta}_s\}_{s=1}^t$ that one uses to estimate the unknown parameter $\theta_\star$. We discuss how to pick the estimator sequence in order to yield a provably small radius of the sets, by using the Follow-the-Regularized-Leader algorithm, which implements a regularized maximum-likelihood estimator. We prove that the radius of the confidence sets is nearly-worst-case optimal, and accordingly, they yield nearly-worst-case regret bounds when used in generalized linear bandit applications. However, due to their data-dependent nature, they can be much tighter than this theory suggests. **Thirdly**, we analyze the limitations of classical (unweighted) LR sets when the underlying conditional observation model is not identifiable. In this case, the resulting (inevitable) estimation bias unnecessarily increases the size of the confidence sets. To mitigate this, we propose an adaptive reweighting scheme that decreases the effect of uninformed early

bias of the estimator sequence on the size of the sets downstream. The reweighting does not affect the coverage guarantees of our sets and utilizes an elegant connection to (robust) powered likelihoods (Wasserman et al., 2020). **Finally**, thanks to the adaptive reweighting scheme, our sets are very practical as we showcase experimentally. We demonstrate that our method works well with exponential and non-exponential family likelihoods, and in parametric as well as in kernelized settings. We attribute their practical benefits to the fact that they *do not depend on (possibly loose) worst-case parameters*.

## 2    The Likelihood Method

The sequential likelihood ratio process (LRP) in (1) is a statistic that compares the likelihood of a given model parameter, with the performance of an adaptively chosen estimator sequence. As noted above, we generalize the traditional definition, which would have $w_s \equiv 1$, and define a corresponding confidence set as

$$\mathcal{C}_t = \{\theta \mid R_t(\theta) \leq 1/\alpha\}. \tag{2}$$

The rationale is that the better a parameter $\theta$ is at explaining the data $\{(x_s, y_s)\}_s^t$ from the true model $\theta_\star$, the smaller this statistic will be, thereby increasing its chances to be included in $\mathcal{C}_t$. When we construct $R_t$, the sequence of $x_s$, $w_s$ and $\hat{\theta}_s$ cannot depend on the noisy observation $y_s$. Formally, consider the filtration $(\mathcal{F}_s)_{s=0}^\infty$ with sub-$\sigma$-algebras $\mathcal{F}_s = \sigma(x_1, \ldots, y_1, \ldots x_s, y_s, x_{s+1})$. We require that $\hat{\theta}_s$ and $w_s$ are $\mathcal{F}_{s-1}$-measurable. Under these very mild assumptions and with arbitrary weights $w_s \in (0, 1]$, we can show coverage, i.e., our (weighted) confidence sets uniformly track the true parameter with probability $1 - \alpha$.

**Theorem 1.** *The stochastic process $R_t(\theta_\star)$ in (1) is a non-negative supermartingale with respect to the filtration $(\mathcal{F}_t)$ and satisfies $R_0(\theta_\star) \equiv 1$. In addition, the sequence $\mathcal{C}_t$ from (2) satisfies $\mathbb{P}_{\theta_\star} (\exists t : \theta_\star \notin \mathcal{C}_t) \leq \alpha$.*

The last statement follows by applying Ville's inequality for super-martingales on $R_t(\theta_\star)$. The proof closely follows Wasserman et al. (2020). While coverage is always guaranteed irrespective of the estimator sequence $\{\hat{\theta}_s\}$, we would like to make the sets as small as possible at fixed coverage, which we do by picking a well-predicting estimator sequence.

### 2.1    The Estimator Sequence Game

The specification of the LR process (LRP) allows us to choose an arbitrary estimator sequence $\{\hat{\theta}_s\}_s$. To understand the importance of the sequence, let us introduce $\theta_\star$ to the definition of $R_t$ in (1), and divide by $\mathcal{L}_t(\{\hat{\theta}_s\}_{s=1}^t)$. This gives the equivalent formulation

$$\mathcal{C}_t := \left\{ \theta \, \middle| \, \frac{\mathcal{L}_t(\theta_\star)}{\mathcal{L}_t(\theta)} \leq \frac{1}{\alpha} \frac{\mathcal{L}_t(\theta_\star)}{\mathcal{L}_t(\{\hat{\theta}_s\}_{s=1}^t)} \leftarrow \text{confidence parameter} \right\}.$$

We see that the predictor sequence does not influence the geometry of the confidence set, which is fully specified by the likelihood function. We also observe that the ratio on the right-hand side serves as a confidence parameter controlling the size (radius) of the confidence sets measured under the likelihood ratio distance to $\theta_\star$. If the confidence parameter goes to zero, only $\theta_\star$ is in the set. The better the estimator sequence is at predicting the data, the smaller the inclusion threshold, and hence the smaller the sets will ultimately be. Specifically, taking the $\log$, we would like to *minimize*

$$\mathcal{R}_t := \log \frac{\mathcal{L}_t(\theta_\star)}{\mathcal{L}_t(\{\hat{\theta}_s\}_{s=1}^t)} = \sum_{s=1}^t -\log(p_{\hat{\theta}_s}^{w_s}(y_s \mid x_s)) - \sum_{s=1}^t -\log(p_{\theta_\star}^{w_s}(y_s \mid x_s)). \tag{3}$$

The quantity $\mathcal{R}_t$ corresponds to a regret in an online prediction game, as will become apparent below.

**Online Prediction Game**    Online optimization is a mature field in interactive learning (Cesa-Bianchi and Lugosi, 2006; Orabona, 2019). The general goal is to minimize a sequence of loss functions as in Eq. (3) and compete against a baseline, which typically is the best-in-hindsight prediction, or – in our case – given by the performance of the fixed parameter $\theta_\star$. Specifically, at every timestep $s$, iteratively, the agent chooses an action $\hat{\theta}_s$ based on $\mathcal{F}_{s-1}$, and a loss function $f_s(\theta)$ is revealed. In most of the online optimization literature, $f_s$ can be chosen adversarially. In our prediction game, we know the whole form of loss function $f_s(\theta) = -\log(p_\theta^{w_s}(y_s \mid x_s))$, as can be seen in (3), and not just $f_s(\hat{\theta}_s)$. Opposed to traditional assumptions in online prediction, in our case, $f_s$ are non-adversarial, but have a stochastic component due to $y_s$. Also, contrary to most instances of online prediction, we do not compare against the best-in-hindsight predictor, but $\theta_\star$ instead, as this is more meaningful in our setting.

**Online Optimization Algorithms**    Generally, we seek an algorithm that incurs low regret. Here, we focus on *Follow-the-Regularized Leader (FTRL)*, which corresponds exactly to using regularized maximum likelihood estimation, making it a natural and computationally practical choice. The update rule is defined in Alg. 1 (Line 3). While other algorithms could be considered, FTRL enjoys the optimal regret rate for generalized linear regression as we show later, and is easily implemented. In order to run the algorithm, one requires a sequence of strongly convex regularizers. For now, let us think of it as $\psi_s(\theta) = \lambda||\theta||_2^2$, which we use in practice. However, one can derive a tighter analysis for a slightly modified, time-dependent regularization strategy for generalized linear models as we show in Sec. 3.3.

## 2.2   Adaptive Reweighting: Choosing the Right Loss

There is yet more freedom in the construction of the LR, via the selection of the *loss function*. Not only do we select the predictor sequence, but also the *weights* of the losses via $w_t$. This idea allows controlling the influence of a particular data point $(x_t, y_t)$ on the cumulative loss based on the value of $x_t$. For example, if we know a priori that for a given $x_t$ our prediction will be most likely bad, we can opt out of using the pair $(x_t, y_t)$ by setting $w_t = 0$. Below we will propose a weighting scheme that depends on a notion of *bias*, which captures how much of the error in predicting $y_t$ is due to our uncertainty about $\hat{\theta}_t$ (compared to the uncertainty we still would have *knowing* $\theta_\star$). Sometimes this *bias* is referred to as *epistemic* uncertainty in the literature, while the residual part of the error is referred to as *aleatoric*. Putting large weight on a data point heavily affected by this bias might unnecessarily increase the regret of our learner (and hence blow up the size of the confidence set). Note that, conveniently, even if we put low weight (zero) on a data point, nothing stops us from using this sample point to improve the estimator sequence in the next prediction round. As we will show below, our reweighting scheme is crucial in defining a practical algorithm for Reproducing Kernel Hilbert Space (RKHS) models and in high signal-to-noise ratio scenarios. Since we do not know $\theta_\star$, our strategy is to compute an *estimate of the bias of the estimator* $\hat{\theta}_t$ and its effect on the value of the likelihood function for a specific $x$ that we played. We use the value of the bias to rescale the loss via $w_t$ such that its effect is of the same magnitude as the statistical error (see Algorithm 1; we call this step BIAS-WEIGHTING).

**Intuition**    To give a bit more intuition, suppose we have a Gaussian likelihood. Then the negative log-likelihood of $(x_t, y_t)$ with weighting is proportional to $\frac{w_t}{\sigma^2}(y_t - x_t^\top \hat{\theta}_t)^2$. Now, if $x_t$ does not lie in the span of the data points $\{x_s\}_{s=1}^{t-1}$ used to compute $\hat{\theta}_t$, it is in general unavoidable to incur large error, inversely proportional to $\sigma^2$. To see this, let us decompose the projection onto $x_t$ as

$$x_t^\top(\hat{\theta}_t - \theta_\star) = \underbrace{x_t^\top(\hat{\theta}_t - \mathbb{E}[\hat{\theta}_t])}_{\text{statistical error}} + \underbrace{x_t^\top(\mathbb{E}[\hat{\theta}_t] - \theta_\star)}_{\text{bias}_{x_t}(\hat{\theta}_t)}, \tag{4}$$

where the first term represents the statistical error up to time $t$, while the second, bias, is deterministic, and independent of the actual realization $y$, depending only $\theta_\star$. Estimators with non-zero bias are *biased*. Plugging this into the likelihood function, we see that in expectation $\frac{1}{\sigma^2}\mathbb{E}[(y_t - x_t^\top \hat{\theta}_t)^2|\mathcal{F}_{t-1}] \lesssim \frac{1}{\sigma^2}\text{bias}_{x_t}^2(\hat{\theta}_t) + \epsilon^2 + \frac{C}{t}$, where $\epsilon^2$ is the unavoidable predictive error in expectation (due to a noisy objective) and is a constant independent of $\sigma^2$. $\frac{C}{t}$ is the statistical error, and $C$ is independent of $\sigma^2$. Note that the bias term scales inversely with the variance, and leads to unnecessarily big confidence parameters for small $\sigma^2$.

In fact, the problem is that we use the likelihood to measure the distance between two parameters, but this is only a "good" distance once the deterministic source of the error (bias) vanishes. For this reason, without weighting, the incurred regret blows up severely in low-noise settings. To counter this, we balance the deterministic estimation bias and noise variance via proper selection of $w_t$. In this case, it turns out that $w_t = \frac{\sigma^2}{\sigma^2 + \text{bias}_{x_t}^2(\hat{\theta}_t)}$ ensures that the overall the scaling is independent of $\sigma^2$. While the choice of weights $\{w_s\}_s^t$ influences the geometry of the confidence sets, with a *good data collection and estimation strategy* the bias asymptotically decreases to zero, and hence the weights converge to $1$.

**Bias estimation**    In order to generalize this rule beyond Gaussian likelihoods, we need a proper generalization of the bias. Our generalization is motivated by our analysis of generalized linear models, but the method can be applied more broadly. The role of the squared statistical error (variance) is played by the inverse of the smoothness constant of the negative log-likelihood functions $f_s$, denoted by $L$. This is the usual smoothness, commonly seen in the convex optimization literature.

We consider penalized likelihood estimators with strongly convex regularizers (Alg. 1, line 3). For this estimator class, we define the bias via a hypothetical stochastic-error-free estimate $\hat{\theta}_t^{\times}$, had we access to the expected values of the gradient loss functions (a.k.a. score). We use the first-order optimality conditions and the indicator function of the set $\Theta$, $i_{\Theta}$, to define the error-free-estimate $\hat{\theta}_t^{\times}$, and the bias of the estimator $\hat{\theta}_t$ as

$$\text{bias}_{x_t}^2(\hat{\theta}_t) = (x_t^{\top}(\theta_{\star} - \hat{\theta}_t^{\times}))^2 \quad \text{with} \quad \mathbb{E}\left[\sum_{s=1}^{t-1} \nabla \log p_{\hat{\theta}_t^{\times}}(y_s|x_s)\right] - \nabla \psi_t(\hat{\theta}_t^{\times}) + i_{\Theta}(\hat{\theta}_t^{\times}) = 0, \quad (5)$$

where the expectation denotes a sequence of expectations conditioned on the prior filtration. This notion of bias coincides with the definition of bias in Eq. (4) for the Gaussian likelihood. This quantity cannot be evaluated in general, however, we prove a computable upper bound.

**Theorem 2** (Bias estimate). *Let the negative log-likelihood have the form,* $-\log p_{\theta}(y_s|x_s) = g(x_s^{\top}\theta)$, *where* $g : \mathbb{R} \to \mathbb{R}$ *is* $\mu$ *strongly-convex and let the regularizer be* $\psi_t(\theta) = \lambda||\theta||_2^2$ *making the overall objective strongly convex. Then, defining* $\mathbf{V}_t^{\mu;\lambda} = \sum_{s=1}^{t} \mu x_s x_s^{\top} + \lambda \mathbf{I}$, *we can bound*

$$\text{bias}_x^2(\hat{\theta}_t) \leq 2\lambda||\theta_{\star}||_2^2 x^{\top}(\mathbf{V}_t^{\mu;\lambda})^{-1}x. \quad (6)$$

The proof is deferred to App. A.3, and requires elementary convex analysis.

This leads us to propose the weighting scheme $w_t = \frac{1/L}{\text{bias}_{x_t}^2(\hat{\theta}_t) + 1/L}$. We justify that this is a sensible choice by analyzing the confidence set on the GLM class in Section 3, which satisfies the smoothness and strong-convexity conditions. We show that this rule properly balances the stochastic and bias components of the error in the regret as in (3). However, this rule is more broadly applicable beyond the canonical representation of GLM or the GLM family altogether.

---

**Algorithm 1** Constructing the LR Confidence Sequence

---

1: **Input:** convex set $\Theta \subset \mathbb{R}^d$, confidence level $\alpha > 0$, likelihood $p_{\theta}(y|x)$, regularizers $\{\psi_t\}_t$
2: **for** $t \in \mathbb{N}_0$ **do**
3:      $\hat{\theta}_t = \arg\min_{\theta \in \Theta} \sum_{s=1}^{t-1} -\log p_{\theta}(y_s \,|\, x_s) + \psi_t(\theta)$          ▷ FTRL
4:      $w_t = \begin{cases} \frac{1/L}{1/L + \text{bias}_{x_t}^2(\hat{\theta}_t)} & \text{THIS WORK} \\ 1 & \text{CLASSICAL} \end{cases}$    ▷ BIAS-WEIGHTING $\text{bias}_{x_t}(\hat{\theta}_t)$ in Eq. (5) or Eq.(6)
5:      $\mathcal{C}_t = \left\{\theta \in \Theta \,\middle|\, \prod_{s=1}^{t} \frac{p_{\hat{\theta}_s}^{w_s}(y_s \,|\, x_s)}{p_{\theta}^{w_s}(y_s \,|\, x_s)} \leq \frac{1}{\alpha}\right\}.$          ▷ Confidence set
6: **end for**

---

## 3 Theory: Linear Models

While the *coverage* (i.e., "correctness") of the likelihood ratio confidence sets is always guaranteed, their worst-case *size* (affecting the "performance") cannot be easily bounded in general. We analyze the size and the geometry of the LR confidence sequence in the special but versatile case of generalized linear models.

### 3.1 Generalized Linear Models

We assume knowledge of the conditional probability model $p_{\theta}(y|x)$, where the covariates $x \in \mathcal{X} \subset \mathbb{R}^d$, and the true underlying model parameter lies in a set $\Theta \subset \mathbb{R}^d$. If $t$ is indexing (discrete) time, then $x_t$ is acquired sequentially, and the – subsequently observed – $y_t$ is sampled from an exponential family distribution parametrized as

$$p_{\theta}(y \,|\, x_t) = h(y) \exp\left(T(y) \cdot x_t^{\top}\theta - A(x_t^{\top}\theta)\right). \quad (7)$$

Here, $A$ is referred to as the *log-partition function* of the conditional distribution, and $T(y)$ is the sufficient statistic. The function $h$ is the base measure, and has little effect on our further developments, as it cancels out in the LR. Examples of commonly used exponential families (Gaussian, Binomial, Poisson or Weibull) with their link functions can be found in Table 1 in App. A.1.

In order to facilitate theoretical analysis for online algorithms, we make the following assumptions about the covariates $x \in \mathcal{X}$ and the set of plausible parameters $\Theta$.

**Assumption 1.** *The covariates are bounded, i.e., $\sup_{x \in \mathcal{X}} \|x\|_2 \leq 1$, and the set $\Theta$ is contained in an $\ell_2$-ball of radius $B$. We will also assume that the log-partition function is strongly convex, that is, that there exists $\mu := \inf_{z \in [-B,B]} A''(z)$, and that $A$ is $L$-smooth, i.e. $L := \sup_{z \in [-B,B]} A''(z)$.*

These assumptions are common in other works addressing the confidence sets of GLMs (Filippi et al., 2010; Faury et al., 2020), who remark that the dependence on $\mu$ is undesirable. However, in contrast to these works, our confidence sets *do not* use these assumptions in the construction of the sets. We only require these for our theoretical analysis. As these are worst-case parameters, the practical performance can be much better for our sets.

### 3.2 Geometry and Concentration

Before stating our results, we need to define a distance notion that the convex negative log-likelihoods induce. For a continuously differentiable convex function $f$, we denote the Bregman divergence as $D_f(a,b) := f(a) - f(b) - \nabla f(b)^\top (a-b)$. The $\nu$-regularized sum of log-partition functions is defined as

$$Z_t^\nu(\theta) := \sum_{s=1}^t w_s A(x_s^\top \theta) + \frac{\nu}{2} \|\theta\|_2^2. \tag{8}$$

This function will capture the geometry of the LR confidence sets. The confidence set size depends mainly on two terms. One refers to a notion of complexity of the space referred to as *Bregman information gain*: $\Gamma_t^\nu(\tilde{\theta}_t) = \log \left( \frac{\int_{\mathbb{R}^d} \exp(-\frac{\nu}{2} \|\theta\|_2^2) d\theta}{\int_{\mathbb{R}^d} \exp(-D_{Z_t^\nu}(\theta, \tilde{\theta}_t)) d\theta} \right)$, first defined by Chowdhury et al. (2022) as a generalization of the *information gain* of Srinivas et al. (2009), $\gamma_t^\nu = \log \left( \det(\sum_{i=1} \frac{\mu}{\nu} x_i x_i^\top + \mathbf{I}) \right)$ for Gaussian likelihoods. We will drop the superscript whenever the regularization is clear from context and simply refer to $\gamma_t$. This term appears because one can relate the decay of the likelihood as a function of the Bregman Divergence from $\theta_\star$ with the performance of a (regularized) maximum likelihood estimator via convex (Fenchel) duality. In particular, if $\tilde{\theta}_t$ is a regularized MLE, $\Gamma_t^\nu := \Gamma_t^\nu(\tilde{\theta}_t)$ will asymptotically scale as $\mathcal{O}(d \log t)$ (cf. Chowdhury et al., 2022, for further discussion). For Gaussian likelihoods and $w_s \equiv 1$, it coincides with the classical information gain independent of $\tilde{\theta}_t$. The second term that affects the size is the regret $\mathcal{R}_t$ of the online prediction game over $t$ rounds we introduced previously in (3). These two parts together yield the following result:

**Theorem 3.** *Let $\nu > 0$ and $\alpha, \delta \in (0,1)$. For the level $1-\alpha$ confidence set $\mathcal{C}_t$ defined in (2) under the GLM in (7), with probability $1 - \delta$, for all $t \geq 1$, any $\theta \in \mathcal{C}_t$ satisfies*

$$D_{Z_t^\nu}(\theta, \theta_\star) \leq \frac{4L}{\mu} \xi_t + 2 \log \left( \frac{1}{\delta} \right) + 2\mathcal{R}_t, \tag{9}$$

*where $\xi_t = \left( \log \left( \frac{1}{\alpha} \right) + \nu B^2 + \Gamma_t^\nu \right)$ and $L, \mu$ are defined as above and finally $\mathcal{R}_t$ is the regret of the game in Eq. (3).*

The set defined via the above divergence does not coincide with the LR confidence set. It is slightly larger due to a term involving $\nu$ (as in Eq. (8)). This is a technical consequence of our proof technique, where the gradient of $Z_t^\nu$ needs to be invertible, and regularization is added to this end. We note that this $\nu > 0$ can be chosen freely. Note that the theorem involves two confidence levels, $\alpha$ and $\delta$: $\alpha$ is a bound on the Type I error – coverage of the confidence sets – while $\delta$ upper bounds the probability of a large radius – and is therefore related to the power and Type II error of a corresponding hypothesis test. The proof of the theorem is deferred to App. B.2.

To give more intuition on these quantities, let us instantiate them for the Gaussian likelihood case with $w_s \equiv 1$. In this scenario, $Z_t^\nu(\theta) = \sum_{s=1}^t \frac{1}{2\sigma^2} \|\theta\|_{x_s x_s^\top}^2 + \frac{\nu}{2} \|\theta\|_2^2$, and the (in this case symmetric) Bregman divergence is equal to $D_{Z_t^\nu}(\theta_\star, \theta) = \frac{1}{2} \|\theta - \theta_\star\|_{\mathbf{V}_t^{\sigma^{-2};\nu}}^2$, where $\mathbf{V}_t^{\mu;\nu} = \sum_{s=1}^t \mu x_s x_s^\top + \nu \mathbf{I}$, which means that our confidence sets are upper bounded by a ball in the same norm as those in the seminal work on linear bandits (Abbasi-Yadkori et al., 2011).

### 3.3 Online Optimization in GLMs: Follow the Regularized Leader

The size of the confidence sets in Theorem 3 depends on the regret of the online prediction game involving the estimator sequence. We now bound this regret when using the Follow-the-Regularized-Leader (FTRL) algorithm in this setting. This high probability bound is novel to the best of our

knowledge and may be of independent interest. We state in a weight-agnostic manner first, and then with our particular choice. The latter variant uses a specifically chosen regularizer. In this case, we can track the contribution of each time-step towards the regret separately.

**Theorem 4.** *Let $\psi_t(\theta) = \lambda ||\theta||_2^2$. Assume Assumption 1, and additionally that $A$ is $L$-smooth everywhere in $\mathbb{R}^d$, and let $w_t \in [0, 1]$ be arbitrary. Then, with probability $1 - \delta$ the regret of FTRL (Alg. 1) satisfies for all $t \geq 1$*

$$\mathcal{R}_t \leq \lambda B^2 + \frac{L}{\mu}(\gamma_t^\lambda + 2\log(1/\delta)) + \frac{2L^2 B^2}{\mu}\gamma_t^\lambda. \tag{10}$$

The regret bounds are optimal in the orders of $\gamma_t^\lambda$, matching lower bounds of Ouhamma et al. (2021), as for linear models $\gamma_t = \mathcal{O}(d \log t)$. Combining results of Thm. 4 with Thm. 3, we get a confidence parameter that scales with $\mathcal{O}(\sqrt{\gamma_t})$, for confidence sets of the form $||\theta - \theta_\star||_{\mathbf{V}_t}$, which coincides with the best-known confidence sets in this setting in the worst-case (Abbasi-Yadkori et al., 2012). The requirement of global $L-$smoothness can be relaxed to $L-$smoothness over $\Theta$. With a more elaborate (but less insightful) analysis, we can show that we achieve a $\tilde{\mathcal{O}}(\gamma_t)$ bound even in this case. The proofs of these results are deferred to App. C.4, App. C.5 and App. C.6 respectively.

**Regret, Weighting and Estimation Bias** Interestingly, the term in Thm. 4 involving the (crude) proxy to the bias – the bound $B$ – is not scaled by the same $L/\mu$ factors as the other terms in the regret bound (10) and in Theorem 3. Namely, the prefactor is $L^2/\mu$ instead of $L/\mu$. This extra dependence manifests itself in the unnecessary penalization through the estimation bias we introduced in Sec. 2.2, particularly in low-noise settings. We addressed this issue by picking the weights $\{w_t\}$. While the above theorem holds for any valid weighting, it does not exhibit the possible improvement from using specific weights.

We argued earlier that the error in prediction should not be measured by the likelihood function if there is deterministic error, since initially, we are fully uncertain about the value of $\theta_\star^\top(\cdot)$ outside the span of previous observations. Of course, if our goal would be to purely pick weights to minimize $\mathcal{R}_t$, then $w_s = 0$ would lead to zero regret and hence be optimal. However, the likelihood ratio would then be constant, and uninformative. In other words, the associated log-partition Bregman divergence in Theorem 3 would be trivial and not filter out any hypotheses. Clearly, some balance has to be met. With this motivation in mind, we proposed a *nonzero* weighting that decreases the regret contribution of the bias, namely $w_t = \frac{1/L}{1/L + \text{bias}_{x_t}^2(\hat{\theta}_t)}$. The advantage of this choice becomes more apparent when we use the regularizer $\psi_t(\theta) = \lambda ||\theta||^2 + A(x_t^\top\theta)$ to obtain the following result.

**Theorem 5.** *Let $\psi_s(\theta) = \lambda ||\theta||^2 + A(x_s^\top\theta)$. Assume Assumption 1, and additionally that $A$ is $L$-smooth everywhere in $\mathbb{R}^d$, and choose $w_s = \frac{1/L}{1/L + \text{bias}_{x_s}(\hat{\theta}_s)^2}$. Additionally, let the sequence of $x_s$ be such that, $\sum_s(1 - w_s)(f_s(\theta_\star) - f_s(\bar{\theta}_{s+1})) \leq L/\mu\gamma_t^\lambda$, where $\bar{\theta}_s$ is the FTRL optimizer with the regularizer $\lambda ||\theta||_2^2$ from Theorem 4 [2]. Then, with probability $1 - \delta$ the regret of FTRL (Alg. 1) satisfies for all $t \geq 1$*

$$\mathcal{R}_t \leq \lambda B^2 + \frac{2L}{\mu}\left(\gamma_t^\lambda + \log\left(\frac{1}{\delta}\right)\right) + \frac{L}{\mu}\sum_{s=1}^t \frac{B^2}{1/L + \text{bias}_{x_s}^2(\hat{\theta}_s)}\Delta\gamma_s^\lambda,$$

*where $\Delta\gamma_s^\lambda = \gamma_{s+1}^\lambda - \gamma_s^\lambda$.*

One can see that for points where the information gain $\Delta\gamma_s$ is large (corresponding to more unexplored regions of the space, where the deterministic source of error is then large), the weighting scheme will make sure that the multiplicative contribution of $B^2$ is mitigated, along with having the correct prefactor $L/\mu$. The reader may wonder how this result is useful when we replace $\text{bias}_{x_s}^2(\hat{\theta}_s)$ with the upper bound from Thm. 2. While instructive, our bound still only makes the bias proxy $B^2$ appear in front of the information gain $\Delta\gamma_t$, instead of the more desireable bias itself. In the latter case, we could also directly make use of the upper bound and get an explicit result only using an upper bound on the bias. We leave this for future work.

We point out that this choice of $\psi_s(\theta)$ in Theorem 5 corresponds to the Vovk-Azoury-Warmuth predictor (Vovk, 2001; Azoury and Warmuth, 1999) in the online learning literature. This choice is helpful in order to track the bias contribution more precisely in our proof.

---

[2]Note that this assumption was missing in an earlier version. $\bar{\theta}_{s+1}$ corresponds to a regularized MLE that *did* observe the data pair $(x_s, y_s)$.

# 4 Application: Linear and Kernelized Bandits

Our main motivation to construct confidence sets is bandit optimization. A prototypical bandit algorithm – the Upper Confidence Bound (UCB) (Auer, 2002) – sequentially chooses covariates $x_s$ in order to maximize the reward $\sum_{s=1}^{t} r_{\theta_\star}(x_s)$, where $r_{\theta_\star}$ is the unknown pay-off function parametrized by $\theta_\star$. UCB chooses the action $x_s$ which maximizes the optimistic estimate of the reward in each round, namely

$$x_s = \arg\max_{x \in \mathcal{X}} \max_{\theta \in \mathcal{C}_{s-1}} r_\theta(x), \tag{11}$$

where $\mathcal{C}_{s-1}$ is some confidence set for $\theta_\star$, and can be constructed with Algorithm 1 from the first $s-1$ data points. An important special case is when $r_{\theta_\star}$ is linear (Abe and Long, 1999) or modelled by a generalized linear model (Filippi et al., 2010). In that case, the inner optimization problem is convex as long as $\mathcal{C}_{s-1}$ is convex. The outer optimization is tractable for finite $\mathcal{X}$. In the applications we consider, our confidence sets are convex, and we easily solve the UCB oracle using convex optimization toolboxes.

**Extension to RKHS**  We introduced the framework of LR confidence sets only for finite-dimensional Euclidean spaces. However, it can be easily extended to Reproducing Kernel Hilbert Spaces (RKHS) (Cucker and Smale, 2002). The definition of the LR process in (1) is still well-posed, but now the sets are subsets of the RKHS, containing functions $f \in \mathcal{H}_k$. An outstanding issue is how to use these sets in downstream applications, and represent them tractably as in Figure 1. Conveniently, even with infinite-dimensional RKHSs, the inner-optimization in (11) admits a Lagrangian formulation, and the generalized representer theorem applies (Schölkopf et al., 2001; Mutný and Krause, 2021). In other words, we can still derive a pointwise upper confidence band as $\mathrm{ucb}(x) = \max_{f \in \mathcal{H}_k, ||f||_k \leq B, f \in C_s} \langle f, k(x, \cdot) \rangle$ in terms of $\{x_j\}_{j=1}^{s} \cup \{x\}$, leading to a $s+1$-dimensional, tractable optimization problem.

We also point out that the weighting is even more paramount in the RKHS setting, as the bias never vanishes for many infinite dimensional Hilbert spaces (Mutný and Krause, 2022). For this purpose, our weighting is of paramount practical importance, as we can see in Figure 2a), where the gray arrow represents the significant improvement from reweighting.

## 4.1 Instantiation of the Theory for Linear Bandits

Before going to the experiments, we instantiate our theoretical results from Sec. 3 to the important and well-studied special case of linear payoffs. In that case, $r_\theta(x) = \langle x, \theta \rangle$ and the agent observes $y_s = \langle x_s, \theta_\star \rangle + \eta_s$ upon playing action $x_s$, where $\eta_s \sim \mathcal{N}(0, \sigma^2)$. We are interested in minimizing the so-called cumulative pseudo-regret, namely, $\mathfrak{R}_t = \sum_{s=1}^{t}[\langle x_\star, \theta_\star \rangle - \langle x_s, \theta_\star \rangle]$, where $x_\star$ refers to the optimal action. Using the set from (2) along with Theorem 3 and the FTRL result of Theorem 4 we can get a regret bound for the choice $w_s \equiv 1$.

**Theorem 6.** *Let $w_s \equiv 1$. For any $\lambda \geq \frac{1}{\sigma^2}$, with probability at least $1 - 3\delta$, for all $t \in \mathbb{N}$ we have*

$$\mathfrak{R}_t \leq 6\sqrt{t\gamma_t^\lambda} \left( \sigma\sqrt{\log(1/\delta) + \gamma_t^\lambda} + \sigma\lambda^{1/2}B + B\sqrt{\gamma_t^\lambda} \right).$$

Our results are optimal in both $d$ and $t$ up to constant and logarithmic factors. The proof is deferred to App. D, but is an instantiation of the aforementioned theorems, along with a standard analysis. There, we also compare to the seminal result of Abbasi-Yadkori et al. (2011), which does not suffer from the dependence on $B\sqrt{\gamma_t}$. We attribute this to the incurred bias in the absence of the reweighting scheme.

For the weighted likelihood ratio, we can obtain a result similar to the above, but multiplied by an upper bound on $\sup_{s \geq 1} w_s^{-1}$. This is undesirable, as our experiments will show that the reweighting scheme vastly improves performance. While this could be somewhat mitigated by using the Theorem 5 instead of Theorem 4 to bound the FTRL regret, a better result should be achievable using our weighting scheme that improves upon Theorem 6 and possibly even matches Abbasi-Yadkori et al. (2011) exactly in the worst-case. We leave this for future work.

## 4.2 Experimental Evaluation

In this subsection, we demonstrate that the practical applicability goes well beyond the Gaussian theoretical result from the previous subsection. In the examples below, we always use the UCB

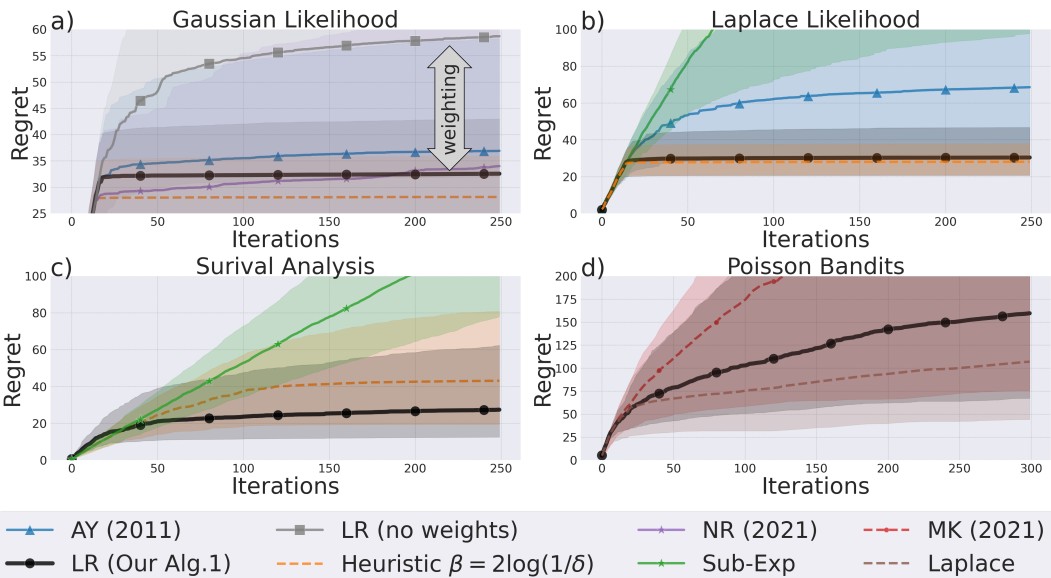

Figure 2: Bandit experiments: On the $y$-axis we report cumulative regret, while the $x$-axis shows the number of iterations. In a) and b) we report the results for linear models with different parametric additive noise. In c) we report the results on a survival analysis with a log-Weibull distribution ($p = 2$) and in d) we showcase Poisson bandits. See App. E for more details. Heuristic methods are *dashed*, while provable are *solid*. Our sets perform the best among all provable methods. Notice in a) the difference in gray and black represents the improvement due to adaptive weighting over $w_s = 1$ for all $s \in [t]$. For each experiment we did 10 reruns, median values are plotted.

algorithm but employ different confidence sets. In particular, we compare our LR confidence sets for different likelihood families with alternatives from the literature, notably classical sub-family confidence sets (Abbasi-Yadkori et al., 2011; Mutný and Krause, 2021), and the robust confidence set of Neiswanger and Ramdas (2021). In practice, however, the radius of these confidence sets is often tuned heuristically. We include such sets as a baseline *without* provable coverage as well. The main take-home message from the experiments is that among all the estimators and confidence sets that enjoy *provable* coverage, our confidence sets perform the best, on par with successful heuristics. For all our numerical experiments in Figure 2, the true payoff function is assumed to be an infinite dimensional RKHS element. For further details and experiments, please refer to App. E.

**Additive Noise Models** Suppose that $r_{\theta_\star}$ is linear and we observe $y_s = x_s^\top \theta_\star + \eta_s$, where $\eta_s$ is additive noise, and $\theta_\star$ is an element of a Hilbert space. We consider classical Gaussian noise as well as Laplace noise in Fig. 2[a), b)]. Notice that in both cases our confidence sets yield lower regret than any other provably valid method. In both cases they are performing as good as *heuristic* confidence sets with confidence parameter $\beta_t \equiv 2 \log(1/\delta)$. The sub-Gaussian confidence sets of Abbasi-Yadkori et al. (2011) (AY 2011) are invalid for the Laplace distribution as it is not sub-Gaussian but only sub-Exponential. For this reason, we compare also with sub-exponential confidence sets derived similarly to those of (Faury et al., 2020). The confidence sets of (Neiswanger and Ramdas, 2021) (NR 2021) perform similarly on Gaussian likelihood, but are only applicable to this setting, as their generalization to other likelihood families involves intractable posterior inference. We note also the difference between the unweighted LR and the weighted one. The examples in Fig. 2 use the true payoff functions $r(x) = -(1.4 - 3x) \sin(18x)$, which we model as an element of a RKHS with squared exponential kernel lengthscale $\gamma = 6 \times 10^{-2}$ on $[0, 1.2]$, which is the baseline function no. 4 in the global optimization benchmark database *infinity77* (Gavana, 2021). Additional experiments can be found in App. E.

**Poisson Bandits** A prominent example of generalized linear bandits (GLB) are Poisson bandits, where the linear payoff is scaled by an exponential function. We instantiate our results on a common benchmark problem, and report the results in Fig. 2d). We improve the regret of UCB for GLBs compared to two alternative confidence sets: one that uses a Laplace approximation with a heuristic confidence parameter, and one inspired by considerations in Mutný and Krause (2021) (MK 2021), also with a heuristic confidence parameter. Note that we cannot compare directly to their provable results in their original form as they do not state them in the canonical form of the exponential family.

**Survival Analysis**   Survival analysis is a branch of statistics with a rich history that models the lifespan of a service or product (Breslow, 1975; Cox, 1997; Kleinbaum and Klein, 2010). The classical approach postulates a well-informed likelihood model. Here, we use a specific hazard model, where the survival time $T$ is distributed with a Weibull distribution, parametrized by $\lambda$ and $p$. The *rate* $\lambda_\theta(x) = \exp(x^\top \theta)$ differs for each configuration $x$, and $p$ – which defines the shape of the survival distribution – is fixed and known. We assume that the unknown part is due to the parameter $\theta$ which is the quantity we build a confidence set around to use within the UCB Algorithm. In particular, the probability density of the Weibull distribution is $P(T = t|x) = \lambda_\theta(x)pt^{p-1}\exp(-t^p\lambda_\theta(x))$. In fact, with $p = 2$, the confidence sets are convex and the UCB rule can be implemented efficiently.

Interestingly, this model admits an alternate linear regression formulation. Namely upon using the transformation $Y = \log T$, the transformed variables $Y|x$ follow a Gumbel-type distribution, with the following likelihood that can be obtained by the change of variables $P(Y = y|x) = \lambda_\theta(x)p\exp(y)^p\exp(-\exp(y)^p\lambda_\theta(x))$. The expectation over $Y$ allows us to express it as a linear regression problem since $\mathbb{E}[Y|x] = -(\theta^\top x + \gamma)/p$, where $\gamma$ is the Euler-Mascheroni constant. More importantly, $Y|x$ is sub-exponential. Hence, this allows us to use confidence sets for sub-exponential variables constructed with the pseudo-maximization technique inspired by Faury et al. (2020). More details on how these sets are derived can be found in App. E. However, these approaches necessarily employ crude worst-case bounds and as can be seen in Figure 2c) the use of our LR-based confidence sequences substantially reduces the regret of the bandit learner.

## 5   Related Work and Conclusion

**Related Work**   The adaptive confidence sequences stem from the seminal work of Robbins et al. (1972), who note that these sets have $\alpha$-bounded Type I error. The likelihood ratio framework has been recently popularized by Wasserman et al. (2020) for likelihood families without known test statistics under the name *universal inference*. This approach, although long established, is surprisingly uncommon in sequential decision-making tasks like bandits. This might be due to the absence of an analysis deriving the size of the confidence sets (Mutný and Krause, 2021), a necessary ingredient to obtain regret bounds. We address this gap for generalized linear models. Another reason might be that practitioners might be interested in non-parametric *sub*-families – a scenario our method does not cover. That being said, many fields such as survival analysis (Cox, 1997) *do* have well-informed likelihoods. However, most importantly, if used naively, this method tends to fail when one departs from assumptions that our probabilistic model is identifiable (i.e., $p_\theta(\cdot \mid x) = p_{\tilde{\theta}}(\cdot \mid x)$ even if $\theta \neq \tilde{\theta}$). We mitigate this problem by introducing the scaling parameters $w_t$ in Eq. (1) to deal with it.

Prevalent constructions of anytime-valid confidence intervals rely on carefully derived concentration results and for a specific estimator such as the least-squares estimator and noise sub-families such as sub-Gaussian, sub-Bernoulli and sub-Poisson Abbasi-Yadkori et al. (2011); Faury et al. (2020); Mutný and Krause (2021). Their constructions involve bounding the suprema of collections of self-normalized stochastic processes (Faury et al., 2020; Mutný and Krause, 2021; Chowdhury et al., 2022). To facilitate closed-form expressions, worst-case parameters are introduced that prohibitively affect the size of the sets – making them much larger than they need to be.

Chowdhury et al. (2022) use the exact form of the likelihood to build confidence sets for parameters of exponential families. However, their approach is restricted to exponential family distributions. They use self-normalization and mixing techniques to explicitly determine the size of the confidence set and do not use an online learning subroutine as we do here. Neiswanger and Ramdas (2021) use likelihood ratios for bandit optimization with possibly misspecified Gaussian processes but is not tractable beyond Gaussian likelihoods. The relation between online convex optimization and confidence sets has been noted in so-called online-to-confidence conversions (Abbasi-Yadkori et al., 2012; Jun et al., 2017; Orabona and Jun, 2021; Zhao et al., 2022), where the existence of a low-regret learner implies a small confidence set. However, these sets still use potentially loose regret bounds to define confidence sets. Our definition is *implicit*. We do not necessarily need a regret bound to run our method, as the radius will depend on the actual, instance-dependent performance of the learner.

**Conclusion**   In this work, we generalized and analyzed sequential likelihood ratio confidence sets for adaptive inference. We showed that with well-specified likelihoods, this procedure gives small, any-time valid confidence sets with model-agnostic and precise coverage. For generalized linear models, we quantitatively analyzed their size and shape. We invite practitioners to explore and use this very versatile and practical methodology for sequential decision-making tasks.

## Acknowledgments and Disclosure of Funding

We thank Wouter Koolen and Aaditya Ramdas for helpful discussions as well as for organizing the SAVI workshop where these discussions took place. NE acknowledges support from the Swiss Study Foundation and the Zeno Karl Schindler Foundation. MM has received funding from the Swiss National Science Foundation through NFP75. This publication was created as part of NCCR Catalysis (grant number 180544), a National Centre of Competence in Research funded by the Swiss National Science Foundation.

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
