# A  Proofs of Theorem 1 and 2

## A.1  GLM Families

Table 1: Examples of exponential family distributions.

| Name | $A(z)$ | $A'(z)$ | $T(y)$ | $\mu$ | $L$ |
|------|--------|---------|--------|-------|-----|
| Gaussian | $z^2/(2\sigma^2)$ | $z/\sigma^2$ | $y/\sigma$ | $1/\sigma^2$ | $1/\sigma^2$ |
| Poisson | $\exp(z)$ | $\exp(z)$ | $y$ | $\exp(-B)$ | $\exp(B)$ |
| Binomial | $\log(1+\exp(z))$ | $\frac{1}{1+\exp(-z)}$ | $y$ | $\mathcal{O}(\exp(-B))$ | $1/4$ |
| Weibull | $k\log(z)-\log k$ | $k/z$ | $y^k$ | $1/B^2$ | $\infty$ |

## A.2  Proof of Theorem 1 (Coverage)

*Proof.* Starting with $\mathbb{E}\left[R_t(\theta_\star)\mid\mathcal{F}_{t-1}\right]$

$$
= \mathbb{E}\left[R_{t-1}(\theta_\star)\frac{p_{\hat\theta_t}^{w_t}(y_t\mid x_t)}{p_{\theta_\star}^{w_t}(y_t\mid x_t)}\;\middle|\;\mathcal{F}_{t-1}\right]
$$

$$
= R_{t-1}(\theta_\star)\int\frac{p_{\hat\theta_t}^{w_t}(y\mid x_t)}{p_{\theta_\star}^{w_t}(y\mid x_t)}p_{\theta_\star}(y\mid x_t)\mathrm{d}y
$$

$$
= R_{t-1}(\theta_\star)e^{((w_t-1)D^r_{w_t}(p_{\theta_\star}(x_t),p_{\hat\theta_t}(x_t)))}\le R_{t-1}(\theta_\star).
$$

The second equality is due to the fact that $R_{t-1}(\theta_\star)$ only depends on $x_1,y_1$ through $x_{t-1},y_{t-1}$. Since $\hat\theta_t$ is $\mathcal{F}_{t-1}$ measurable by assumption, $p_{\hat\theta_t}$ is a density, and if $w_t<1$, the integral is equal to an exponential of the Rényi-divergence $D^r_{w_t}(\cdot,\cdot)$. The negativity of the exponent follows from $w_t<1$ and the non-negativity of the divergence. Note that in the "degenerate" case of $w_t=1$, we can easily see that the integral is over a density (cancellation), and hence also bounded by 1. The last part of the statement follows easily by using Ville's inequality for supermartingales. $\qquad\square$

All the elements of the above proof appear in Wasserman et al. (2020) albeit separately, and not with time-varying powered robust likelihoods.

## A.3  Proof of Theorem 2 (Bias)

We will need a gradient characterization of strong-convexity, which we prove in the following lemma.

**Lemma 1** (Convexity: Gradient). *Defining*

$$
F_t(\theta) = -\sum_{s=1}^{t}\underset{\theta_\star}{\mathbb{E}}[\nabla\log_\theta p(y_s\mid x_s)\mid\mathcal{F}_{s-1}],
$$

*under the assumption $p_\theta(y_s|x_s) = -g(x_s^\top\theta)$ and $g$ is $\mu$-strongly convex, we have for any $\theta\in\Theta$:*

$$
(F_t(\theta) - F_t(\theta_\star))^\top(\theta-\theta_\star) \ge ||\theta-\theta_\star||^2_{\mathbf{V}_t^{\mu;0}}.
$$

*Proof.* We assume that $g$ is $\mu$-strongly convex. Therefore, for any $s\le t$, we get the two inequalities

$$
g(x_s^\top\theta) - g(x_s^\top\theta_\star) \ge g'(x_s^\top\theta_\star)(x_s^\top\theta - x_s^\top\theta_\star) + \frac{\mu}{2}||x_s^\top\theta_\star - x_s^\top\theta||_2^2
$$

$$
g(x_s^\top\theta_\star) - g(x_s^\top\theta) \ge g'(x_s^\top\theta)(x_s^\top\theta_\star - x_s^\top\theta) + \frac{\mu}{2}||x_s^\top\theta_\star - x_s^\top\theta||_2^2.
$$

Adding these two together, we obtain

$$
0 \ge (g'(x_s^\top\theta_\star) - g'(x_s^\top\theta))(x_s^\top(\theta-\theta_\star)) + \mu||\theta-\theta_\star||^2_{x_sx_s^\top}.
$$

Observing that $-\nabla_\theta p_\theta(y_s\mid x_s) = -g'(x_s^\top\theta)x_s$, we can equivalently write

$$
0 \ge (\nabla\log p_\theta(y_s\mid x_s) - \nabla\log p_{\theta_\star}(y_s\mid x_s))^\top(\theta-\theta_\star)) + \mu||\theta-\theta_\star||^2_{x_sx_s^\top}.
$$

This holds for any realization of $y_s$, and hence taking expectations yields

$$(\mathbb{E}[\nabla \log p_{\theta_\star}(y_s \,|\, x_s) \,|\, \mathcal{F}_{s-1}] - \mathbb{E}[\nabla \log p_\theta(y_s \,|\, x_s) \,|\, \mathcal{F}_{s-1}])^\top (\theta - \theta_\star) \geq \mu ||\theta - \theta_\star||^2_{x_s x_s^\top}.$$

Summing up over $s \leq t$ and using the definition of $F_t$ we get

$$(F_t(\theta) - F_t(\theta_\star))^\top (\theta - \theta_\star) \geq ||\theta - \theta_\star||^2_{\mathbf{V}_t^\mu;0}.$$

$\square$

Notice that the estimator in Alg. 1 has to fulfil the KKT conditions. We will denote the condition for belonging to the set as $h(\theta) \leq B^2$, where $h$ is a squared and twice-differentiable norm (there are many choices beyond $|| \cdot ||_2^2$). The KKT conditions are

$$\sum_{s=1}^{t} -\nabla_\theta \log p_\theta(y_s|x_s) + \nabla \psi_t(\theta) + l\nabla h(\theta) = 0 \tag{12}$$

$$l(h(\theta) - B^2) = 0$$
$$l \geq 0,$$

where the second and third conditions represent a complementary slackness requirement. Notice that the system of these equations has a unique solution due to the strong-convexity of the objective, and has to attain a unique minimum on a compact convex subset of $\mathbb{R}^d$. Adding the same quantity on both sides of (12) yields

$$\sum_{s=1}^{t} -\mathbb{E}_{\theta_\star}[\nabla_\theta \log p_\theta(y_s|x_s)|\mathcal{F}_{s-1}] + \nabla \psi_t(\theta) + l\nabla h(\theta)$$

$$= \underbrace{\sum_{s=1}^{t} [\nabla_\theta \log p_\theta(y_s|x_s) - \mathbb{E}_{\theta_\star}[\nabla_\theta \log p_\theta(y_s|x_s)|\mathcal{F}_{s-1}]]}_{:=E_t}. \tag{13}$$

This line motivates the definition of the error-free estimator in $\theta_t^\times$ (6), where $E_t$ is set to zero. We will also make use of a fundamental property of the score (gradient of log-likelihood), namely

$$\mathbb{E}_{y_t \sim p_{\theta_\star}(\cdot \,|\, x)}[\nabla \log p_{\theta_\star}(y_t|x_t)|\mathcal{F}_{t-1}] = 0. \tag{14}$$

A classical textbook reference for this is e.g. McCullagh (2018) but any other classical statistics textbook should contain it. Using these observations, we can already prove Theorem 2.

*Proof of Theorem 2.* Using the optimality conditions of $\theta_t^\times$, $h(\theta) = ||\theta||_2^2$ and $\psi_t(\theta) = ||\theta||_2^2$, we obtain the following statements:

$$\sum_{s=1}^{t} -\mathbb{E}_{\theta_\star}[\nabla \log p_{\theta_t^\times}(y_s|x_s)|\mathcal{F}_{s-1}] + \lambda \theta_t^\times + 2l\theta_t^\times = 0$$

$$\implies \sum_{s=1}^{t} -\mathbb{E}_{\theta_\star}[\nabla \log p_{\theta_t^\times}(y_s|x_s)|\mathcal{F}_{s-1}] + \mathbb{E}_{\theta_\star}[\nabla \log p_{\theta_\star}(y_s|x_s)|\mathcal{F}_{s-1}] + \lambda \theta_t^\times + 2l\theta_t^\times = 0,$$

where in the last line we used the property (14). Now, notice that since we know $\theta_\star$ is generating the data, the best possible explanation without enforcing the constraint and the regularization would be to set $\theta_t^\times = \theta_\star$ as the cross-entropy is minimized at this point, and the above is just the optimality condition for optimizing the cross-entropy between these two distribution. Of course, this is only in the absence of regularization or constraints i.e. $\lambda = 0$. Now, with the regularization constraint, as the true $\theta_\star$ lies inside the constraint $h(\theta) \leq B^2$, and both the regularization and constraints induce star-shaped sets, their effect is to make $\theta_t^\times$ smaller in norm than $\theta^*$. This holds generally for any $h$ which is a norm. As a consequence of this consideration, $||\theta_t^\times||_2 < B$, and then the complementary slackness dictates that $l = 0$.

We can therefore proceed with this simplification. Let us use the shorthand $F_t(\theta) = \sum_{s=1}^{t} -\mathbb{E}_{\theta_\star}[\nabla \log p_\theta(y_s|x_s)|\mathcal{F}_{s-1}]$ and compute

$$F_t(\theta_t^\times) - F(\theta_\star) + \lambda(\theta_t^\times - \theta_\star) \qquad\qquad = -\lambda\theta_\star$$

$$\implies (\theta_t^\times - \theta_\star)^\top (F_t(\theta_t^\times) - F(\theta_\star) + \lambda(\theta_t^\times - \theta_\star)) \qquad = -\lambda(\theta_t^\times - \theta_\star)^\top \theta_\star$$

$$\overset{\text{Lemma } 1}{\implies} ||\theta_t^\times - \theta_\star||^2_{\mathbf{V}_t^{\mu,\lambda}} \qquad\qquad\qquad\qquad \leq -\lambda(\theta_t^\times - \theta_\star)^\top \theta_\star. \qquad (15)$$

It suffices to apply the Cauchy-Schwarz Inequality and invoke (15):

$$
\begin{aligned}
\text{bias}_{x_s}(\hat{\theta}_s)^2 &= (x_s^\top(\hat{\theta}_t^\times - \theta_\star))^2 \\
&\leq ||x_s||^2_{(\mathbf{V}_t^{\mu,\lambda})^{-1}}||\theta_t^\times - \theta_\star||^2_{\mathbf{V}_t^{\mu,\lambda}} \\
&\leq \lambda||x_s||^2_{(\mathbf{V}_t^{\mu,\lambda})^{-1}}\lambda(\theta_t^\times - \theta_\star)^\top(-\theta_\star) \\
&\leq \lambda||x_s||^2_{(\mathbf{V}_t^{\mu,\lambda})^{-1}}||(\theta_t^\times - \theta_\star)||_2||\theta_\star||_2 \\
&\leq 2\lambda||x_s||^2_{(\mathbf{V}_t^{\mu,\lambda})^{-1}}||\theta_\star||^2_2,
\end{aligned}
$$

where in the last inequality we used $||\theta_t^\times||_2 \leq ||\theta_\star||_2$, due to the regularizer, as explained above. $\square$

**GLM models** Let us define the processes

$$S_t = \sum_{s=1}^t x_s T(y_s) \quad \text{and} \quad W_t = \sum_{s=1}^t x_s A'(x_s^\top \theta_\star).$$

In this scenario, an equivalent of (13) then involves the gradient of the regularized (unweighted) log-partition function $Z_t^\lambda$ we defined in (8) and is equal to

$$\sum_{s=1}^t A'(x_s^\top \theta)x_s + \nabla\psi_t(\theta) + l\nabla h(\theta) = \tilde{E}_t, \qquad (16)$$

where $\tilde{E}_t = S_t$ for $\hat{\theta}_t$ and $\tilde{E}_t = W_t$ for $\theta_t^\times$.

# B  Proof of Theorem 3 (Bregman Ball Confidence Set)

**Proof sketch** We give a quick sketch of the proof. To bound the size of the sets, we will draw inspiration from the i.i.d. parameter estimation analysis of Wasserman et al. (2020) and separate out the likelihood ratio in a part that relates the true parameter with the estimator sequence (i.e. regret), and a part that is independent of the estimator and characterized by a supremum of a stochastic process. We want to show that *any* point which is far away from the true parameter will eventually not be included in the confidence set anymore. Defining $\mathcal{L}^{(t)}(\{\hat{\theta}_s\}_{s=1}^t)$ as $\prod_{i=1}^t p_{\hat{\theta}_i}^{w_i}(y_i \,|\, x_i)$, we wish to show that for any $\theta$ far from $\theta_\star$, we have

$$\log\frac{1}{R_t(\theta)} = \log\frac{\mathcal{L}^{(t)}(\theta)}{\mathcal{L}^{(t)}(\theta_\star)} + \log\frac{\mathcal{L}^{(t)}(\theta_\star)}{\mathcal{L}^{(t)}(\{\hat{\theta}_s\}_{s=1}^t)} \leq \log(\alpha),$$

which is equivalent to saying that $\theta \notin \mathcal{C}_t$. The second term corresponds to our notion of regret exactly ($\mathcal{R}_t$, as discussed above). The first term is what we will focus on. We will bound the supremum of $\log\frac{\mathcal{L}^{(t)}(\theta)}{\mathcal{L}^{(t)}(\theta_\star)}$ for all $\theta$ sufficiently far away from $\theta_\star$. "Far away" will be measured in the Bregman divergence outlined above. Note that this quantity can be expected to be negative, in general, (especially for "implausible" parameters), since with enough data, $\theta_\star$ should appear much more likely. Writing this ratio out, we will observe that it is equal to

$$-D_{Z_t^0}(\theta, \theta_\star) + \underbrace{\langle \theta - \theta_\star, \sum_{s=1}^t w_s x_s(T(y_s) - \mathbb{E}_{\theta_\star}[T(y_s)])\rangle}_{\approx \tilde{S}_t}.$$

At this point, it will be sufficient to bound the cross term (second term) over the whole of $\Theta$. We view this supremum as part of the Legendre Fenchel transform of the function $\mathcal{B}_t(\lambda) = D_{Z_t^\nu}(\theta_\star + \lambda, \theta_\star)$:

$$\sup_{\lambda \in \mathbb{R}^d} \left(\lambda^\top \tilde{S}_t - \mathcal{B}_t(\lambda)\right) = (\mathcal{B}_t)^\star(\tilde{S}_t)$$

and harness duality properties of the Bregman divergence, along with known concentration arguments (Chowdhury et al., 2022, Theorem A.1).

### B.1 Technical Lemmas

We need to introduce the concept of Legendre functions:

**Definition 1.** *Let $f : \mathbb{R}^d \to \mathbb{R}$ be a convex function and $C = \text{int}(\text{dom}(f))$. Then, a function is called Legendre if it satisfies*

1. *$C$ is non-empty.*

2. *$f$ is differentiable and strictly convex on $C$.*

3. *$\lim_{n \to \infty} ||\nabla f(x_n)|| = \infty$ for any sequence $(x_n)_n$ with $x_n \in C$ for all $n$ and $\lim_{n \to \infty} x_n = x$ for some $x \in \partial C$.*

This means that the gradient has to blow up near the edge of the domain. Note as well that the boundary condition is vacuous if there is no boundary. Legendre functions have some nice properties, most importantly regarding the bijectivity of their gradients (see e.g. Lattimore and Szepesvári (2020)):

**Lemma 2.** *For a Legendre function $f : \mathbb{R}^d \to \mathbb{R}$*

1. *$\nabla f$ is a bijection between $\text{int}(\text{dom}(f))$ and $\text{int}(\text{dom}(f^*))$ with the inverse $(\nabla f)^{-1} = \nabla f^*$.*

2. *$D_f(x, y) = D_{f^*}(\nabla f(y), \nabla f(x))$ for all $x, y \in \text{int}(\text{dom}(f))$.*

3. *The Fenchel conjugate $f^*$ is also Legendre.*

With this, we can prove a slightly extended result, that appears as Lemma 2.1. in Chowdhury et al. (2022).

**Lemma 3.** *For a Legendre function $f$ we have the identity*

$$D_f(x, y) = (D_{f,x})^*(\nabla f(y) - \nabla f(x))$$

*where we define $D_{f,x}(\lambda) = D_f(x + \lambda, x)$.*

**Notational Shorthands**   Remember the model (7), with log-partition function $A$. We define $A_s(\theta) = w_s A(x_s^\top \theta)$ and $T_s(y) := w_s x_s T(y)$ to denote the log-partition function and the response function of the same exponential family distribution, but parametrized by $\theta$ instead of $x_s^\top \theta$. That this is a valid parametrization can easily be seen from the likelihood definition. Indeed, denote by $p_\beta^{EF}$ the exponential family reward distribution with parameter $\beta$. Then our model (7) can be seen to satisfy

$$p_\theta(y \mid x_s) = p_{x_s^\top \theta}(y) = h(y) \exp(T(y)x_s^\top \theta - A(x_s^\top \theta)) = h(y) \exp(T_s(y)^\top \theta - A_s(\theta)).$$

Exponentiating the likelihood with a weighting $w_s$ gives rise to another exponential family distribution. We can see that

$$p_\theta^{w_s}(y \mid x_s) = h^{w_s}(y) \exp(w_s T(y)x_s^\top \theta - w_s A(x_s^\top \theta)) = h^{w_s}(y) \exp(T_s(y)^\top \theta - A_s(\theta)).$$

Note that this does not necessarily integrate to one, but it is easy to see that there is a normalization function $\tilde{h}$ that makes it integrate to one. Therefore, the following is a valid parametrization of an exponential family distribution:

$$\tilde{h}(y) \exp(T_s(y)^\top \theta - A_s(\theta)).$$

Additionally, let $A_0(\theta) = \frac{\nu}{2}||\theta||_2^2$ be defined on $\mathbb{R}^d$ (i.e. a Legendre Function). We will also define the estimator

$$\tilde{\theta}_t = (\nabla Z_t^\nu)^{-1} \left( \sum_{s=1}^{t} T_s(y_s) \right).$$

This is a well-defined quantity because the gradient will be invertible, by Lemma 2 above.

Conveniently, Chowdhury et al. (2022) prove the following Theorem 7 using an elegant application of the method of mixtures.

**Proposition 1** (Theorem 7 in Chowdhury et al. (2022)). *With probability $1 - \delta$, for all $t \in \mathbb{N}$*

$$D_{Z_t^\nu}(\theta_\star, \tilde{\theta}_t) \le \log(1/\delta) + A_0(\theta_\star) + \Gamma_t^\nu,$$

*where*

$$\Gamma_t^\nu = \log\left(\frac{\int_{\mathbb{R}^d} \exp(-\frac{1}{2}\|\theta\|_2^2)\mathrm{d}\theta}{\int_{\mathbb{R}^d} \exp(-D_{Z_t^\nu}(\theta, \tilde{\theta}_t))\mathrm{d}\theta}\right).$$

Lastly, we will need the (one-argument) function

$$\mathcal{B}_t(\lambda) = D_{Z_t^\nu}(\theta + \lambda, \theta),$$

i.e. a shortcut for the Bregman divergence of $Z_t^\nu$ at $\theta$. We use this one-argument function as we will be interested in its dual. We will also need a lemma on the sub-homogeneity properties of this object.

**Lemma 4.** *Under Assumption 1, for $\theta \in \Theta$ and $\lambda$ such that $\theta + \lambda \in \Theta$, we have for any $\gamma \le \frac{\mu}{2L}$*

$$\mathcal{B}_t(\gamma\lambda) \le \frac{1}{2}\gamma\mathcal{B}_t(\lambda),$$

*i.e. function $g(\gamma) = \mathcal{B}_t(\gamma\lambda)$ is sub-homogeneous with contraction parameter $\frac{1}{2}$ on $[0, \frac{\mu}{2L}]$.*

See Appendix B.3 for a proof.

## B.2 Proof of Theorem 3

As mentioned in the main paper, our proof will show that all $\theta$ sufficiently far from $\theta_\star$ will be excluded from $\mathcal{C}_t$ eventually. Equation (B) in the main text specifies the exclusion criterion, i.e. $\theta \notin \mathcal{C}_t$ if and only if

$$\frac{1}{R_t(\theta)} = \log\frac{\mathcal{L}^{(t)}(\theta)}{\mathcal{L}^{(t)}(\theta_\star)} + \log\frac{\mathcal{L}^{(t)}(\theta_\star)}{\mathcal{L}^{(t)}(\{\hat{\theta}_s\}_{s=1}^t)} \le \log(\alpha). \tag{17}$$

The second term is bounded by the regret of the online learner. And therefore, a sufficient condition for $\theta \notin \mathcal{C}_t$ is

$$\log\left(\frac{\mathcal{L}^{(t)}(\theta)}{\mathcal{L}^{(t)}(\theta_\star)}\right) \le \log(\alpha) - \mathcal{R}_t.$$

Henceforth, we will be interested in having an explicit set $\tilde{\mathcal{C}}_t$ such that we can upper bound

$$\sup_{\theta \notin \tilde{\mathcal{C}}_t} \log\left(\frac{\mathcal{L}^{(t)}(\theta)}{\mathcal{L}^{(t)}(\theta_\star)}\right). \tag{18}$$

This will imply that that $\tilde{\mathcal{C}}_t^c \subset \mathcal{C}_t^c$, or in other words, $\mathcal{C}_t \subset \tilde{\mathcal{C}}_t$. Without further ado, let us derive a more convenient form of the ratio in question

$$
\begin{aligned}
\log\left(\frac{\mathcal{L}^{(t)}(\theta)}{\mathcal{L}^{(t)}(\theta_\star)}\right) &= \log\left(\frac{\prod_{s=1}^t h(y_s)\exp\left(w_s x_s^\top \theta T(y_s) - w_s A(x_s^\top \theta)\right)}{\prod_{s=1}^t h(y_s)\exp\left(w_s x_s^\top \theta T(y_s) - w_s A(x_s^\top \theta_\star)\right)}\right) \\
&= \sum_{s=1}^t w_s x_s^\top \theta T(y_s) - w_s A(x_s^\top \theta) - w_s x_s^\top \theta_\star T(y_s) + w_s A(x_s^\top \theta_\star) \\
&= \sum_{s=1}^t \langle\theta - \theta_\star, w_s T(y_s)x_s\rangle + w_s A(x_s^\top \theta_\star) - w_s A(x_s^\top \theta) \\
&= \sum_{s=1}^t \langle\theta - \theta_\star, w_s T(y_s)x_s\rangle - \left(w_s A(x_s^\top \theta) - w_s A(x_s^\top \theta_\star) - x_s^\top(\theta - \theta_\star)w_s A'(x_s^\top \theta_\star)\right. \\
&\quad \left. + x_s^\top(\theta - \theta_\star)w_s A'(x_s^\top \theta_\star)\right) \\
&= \sum_{s=1}^t \langle\theta - \theta_\star, w_s T(y_s)x_s\rangle - w_s D_A(x_s^\top \theta, x_s^\top \theta_\star) - x_s^\top(\theta - \theta_\star)w_s A'(x_s^\top \theta_\star)
\end{aligned}
$$

$$= -\sum_{s=1}^{t} w_s D_A(x_s^\top \theta, x_s^\top \theta_\star) + \sum_{s=1}^{t} \langle \theta - \theta_\star, \, w_s T(y_s) x_s - x_s w_s A'(x_s^\top \theta_\star) \rangle.$$

We can switch parametrizations as described above:

$$\log\left(\frac{\mathcal{L}^{(t)}(\theta)}{\mathcal{L}^{(t)}(\theta_\star)}\right) = -D_{Z_t^0}(\theta, \theta_\star) + \sum_{s=1}^{t} \langle \theta - \theta_\star, \, T_s(y_s) - \nabla A_s(\theta_\star) \rangle$$

$$= -D_{Z_t^0}(\theta, \theta_\star) + \sum_{s=1}^{t} \langle \theta - \theta_\star, \, T_s(y_s) - \mathbb{E}_{\theta_\star}[T_s(y_s)] \rangle$$

$$= -D_{Z_t^0}(\theta, \theta_\star) + \langle \theta - \theta_\star, \, S_t \rangle, \tag{19}$$

where we define $S_t := \sum_{s=1}^{t} (T_s(y_s) - \mathbb{E}_{\theta_\star}[T_s(y_s)])$.

$Z_t^\nu$ is strictly convex whenever $\nu \neq 0$, and convex otherwise (it might also be strictly convex otherwise, corresponding to some cases where the $x_s$ span the full $d$-dimensional Euclidean space and $w_s > 0$, which will be satisfied uniformly. We note that since $\mathrm{dom}(Z_t^\nu) = \mathbb{R}^d$, $Z_t^\nu$ is, therefore, Legendre, and its gradient is invertible. We will relate our problem to this estimator via the duality properties developed above. First, note that by the well-known fact $\mathbb{E}_{\theta_\star}[T_s(y_s)] = \nabla A_s(\theta_\star)$ and by the definition of $\tilde\theta_t$, we have

$$S_t = \nabla Z_t^\nu \left( (\nabla Z_t^\nu)^{-1} \left( \sum_{s=1}^{t} T_s(y_s) \right) \right) - \nabla Z_t^0(\theta_\star)$$

$$= \underbrace{\nabla Z_t^\nu \left( \tilde\theta_t \right) - \nabla Z_t^\nu(\theta_\star)}_{=:\tilde S_t} + \underbrace{\nabla A_0(\theta_\star)}_{=\nu\theta_\star}. \tag{20}$$

Now, we leverage the duality properties: We can write

$$\sup_{\lambda \in \mathbb{R}^d} \left( \lambda^\top \tilde S_t - \mathcal{B}_t(\lambda) \right) \overset{(i)}{=} (\mathcal{B}_t)^\star (\tilde S_t)$$

$$\overset{(20)}{=} (\mathcal{B}_t)^\star \left( \nabla Z_t^\nu(\tilde\theta_t) - \nabla Z_t^\nu(\theta_\star) \right)$$

$$\overset{\text{Lemma } 3}{=} D_{Z_t^\nu}(\theta_\star, \tilde\theta_t), \tag{21}$$

where $(i)$ is simply the definition of the Legendre-Fenchel transform. Why did we do all this work? Well, we are interested in the supremum in Equation (18). It is sufficient to bound the supremum over all $\theta \in \Theta$ of terms of the form (see Equation (19))

$$\langle \theta - \theta_\star, \, S_t \rangle.$$

While we could do a covering type argument (carefully relaxing the i.i.d. data assumptions typical in empirical process theory), it is much easier to relate this supremum to the estimator via duality.

With probability at least $1 - \delta$, Proposition 1 gives us a high-probability time-uniform bound on

$$D_{Z_t^\nu}(\theta_\star, \tilde\theta_t) \leq \log(1/\delta) + A_0(\theta_\star) + \Gamma_t^\nu,$$

and therefore, by plugging into Equation (21) and making the reparametrization $\lambda = \gamma(\theta - \theta_\star)$ for some positive $\gamma$, it gives us

$$\forall t \geq 0 \; \forall \theta \in \mathbb{R}^d \; \forall \gamma \in \mathbb{R}_+ \; : \; \gamma \tilde S_t^\top (\theta - \theta_\star) - \mathcal{B}_t(\gamma(\theta - \theta_\star)) \leq \log(1/\delta) + A_0(\theta_\star) + \Gamma_t^\nu.$$

Therefore, for all $t \geq 0$ and all $\theta \in \mathbb{R}^d$, the following holds:

$$S_t^\top (\theta - \theta_\star) = \tilde S_t^\top (\theta - \theta_\star) + \nabla A_0(\theta_\star)^\top (\theta - \theta_\star)$$

$$\leq \frac{1}{\gamma} \log(1/\delta) + \frac{1}{\gamma} A_0(\theta_\star) + \frac{1}{\gamma} \Gamma_t^\nu + \frac{1}{\gamma} \mathcal{B}_t(\gamma(\theta - \theta_\star)) + \nabla A_0(\theta_\star)^\top (\theta - \theta_\star).$$

Since $A_0(\theta) = \frac{\nu}{2} \|\theta\|_2^2$, restricting our uniform bound over $\theta \in \Theta$ gives us $\forall t \geq 0 \; \forall \theta \in \Theta$:

$$S_t^\top (\theta - \theta_\star) \leq \frac{1}{\gamma} \log(1/\delta) + \frac{\nu}{2\gamma} B^2 + \frac{1}{\gamma} \Gamma_t^\nu + \frac{1}{\gamma} \mathcal{B}_t(\gamma(\theta - \theta_\star)) + \nu B^2.$$

Now, we note that under Assumption 1, Lemma 4 kicks in and we have for any $t \geq 0, \theta \in \Theta$ and $\gamma = \frac{\mu}{2L}$

$$S_t^\top (\theta - \theta_\star) \leq \frac{1}{\gamma} \log(1/\delta) + \frac{\nu}{2\gamma} B^2 + \frac{1}{\gamma} \Gamma_t^\nu + \frac{1}{2} \mathcal{B}_t(\theta - \theta_\star) + \nu B^2. \tag{22}$$

Finally, we can use this in (19) to obtain

$$
\begin{aligned}
\log\left(\frac{\mathcal{L}^{(t)}(\theta)}{\mathcal{L}^{(t)}(\theta_\star)}\right) &\leq -\mathcal{B}_t(\theta - \theta_\star) + \langle \theta - \theta_\star, \, S_t \rangle \\
&\overset{(22)}{\leq} -\frac{1}{2} \mathcal{B}_t(\theta - \theta_\star) + \frac{1}{\gamma} \log(1/\delta) + \frac{\nu}{2\gamma} B^2 + \frac{1}{\gamma} \Gamma_t^\nu + \nu B^2 \\
&\leq -\frac{1}{2} \mathcal{B}_t(\theta - \theta_\star) + \frac{2L}{\mu}\left(\log(1/\delta) + \frac{\nu B^2}{2} + \Gamma_t^\nu\right) + \nu B^2. \tag{23}
\end{aligned}
$$

It remains to investigate the full likelihood ratio in (17):

$$
\frac{1}{R_t(\theta)} - \log(\alpha)
$$

$$
= \log \frac{\mathcal{L}^{(t)}(\theta)}{\mathcal{L}^{(t)}(\theta_\star)} + \log \frac{\mathcal{L}^{(t)}(\theta_\star)}{\mathcal{L}^{(t)}(\{\hat\theta_s\}_{s=1}^t)} + \log(1/\alpha)
$$

$$
\overset{(23)\,\&\,(3)}{\leq} -\frac{1}{2} \mathcal{B}_t(\theta - \theta_\star) + \frac{2L}{\mu}\left(\log(1/\delta) + \frac{\nu B^2}{2} + \Gamma_t^\nu\right) + \nu B^2 + \log(1/\alpha) + \mathcal{R}_t. \tag{24}
$$

Note that crucially for $\theta \in \Theta$, we have

$$
\theta \notin \mathcal{C}_t \iff \frac{1}{R_t(\theta)} - \log(\alpha) \leq 0.
$$

This is implied by

$$
\mathcal{B}_{Z_t^\nu}(\theta, \theta_\star) \geq \frac{4L}{\mu}\left(\log(1/\delta) + \frac{\nu B^2}{2} + \Gamma_t^\nu\right) + 2\nu B^2 + 2\log(1/\alpha) + 2\mathcal{R}_t,
$$

or, since $L \geq \mu$, more compactly by

$$
\mathcal{B}_{Z_t^\nu}(\theta, \theta_\star) \geq \frac{4L}{\mu}\left(\log(1/\delta) + \nu B^2 + \Gamma_t^\nu\right) + 2\log(1/\alpha) + 2\mathcal{R}_t.
$$

### B.3    Proof of Technical Lemmas

First, we will prove Lemma 3. The proof exactly follows Chowdhury et al. (2022), we include it here for convenience because it is very short.

*Proof.* By definition

$$
\begin{aligned}
&(D_{f,x})^*(\nabla f(y) - \nabla f(x)) \\
&= \sup_{a \in \mathbb{R}^d} \left( \langle a, \, \nabla f(y) - \nabla f(x) \rangle - D_{f,x}(a) \right) \\
&= \sup_{a \in \mathbb{R}^d} \left( \langle a, \, \nabla f(y) - \nabla f(x) \rangle - D_f(x + a, x) \right) \\
&= \sup_{a \in \mathbb{R}^d} \left( \langle a, \, \nabla f(y) - \nabla f(x) \rangle - f(x + a) + f(x) + \langle \nabla f(x), \, a \rangle \right) \\
&= \sup_{a \in \mathbb{R}^d} \left( \langle a, \, \nabla f(y) \rangle - f(x + a) + f(x) \right).
\end{aligned}
$$

Since $f$ is strictly convex and differentiable, first-order optimality conditions imply that the optimal $a$ satisfies $\nabla f(y) - \nabla f(x + a) = 0$ ($a$ is unconstrained). Since the gradient is invertible, we must have $a = y - x$. If we plug this into the above, we have

$$
\begin{aligned}
(D_{f,x})^*(\nabla f(y) - \nabla f(x)) &= \langle y - x, \, \nabla f(y) \rangle - f(y) + f(x) \\
&= f(x) - f(y) - \langle \nabla f(y), \, x - y \rangle \\
&= D_f(x, y).
\end{aligned}
$$

$\square$

Now we prove Lemma 4. To this end, we will do a reduction to the one-dimensional case, and prove the one-dimensional result below.

**Lemma 5.** *Under Assumption 1, for any $a \in [B, B]$, any $\gamma \in (0, \frac{\mu}{2L}]$ and any $\Delta$ with $a + \Delta \in [B, B]$*

$$A(a + \gamma\Delta) - A(a) - A'(a)\gamma\Delta \le \frac{1}{2}\gamma\left[A(a + \Delta) - A(a) - A'(a)\Delta\right].$$

We prove that this implies the desired sublinearity of the full Bregman difference.

*Proof.* (of Lemma 4). Let $\theta$ and $\lambda$ be such that $\theta, \theta + \lambda \in \Theta$. We will first show that for any $s \in \{0, \dots, t\}$, $B_{A_s(\theta, \theta + \cdot)}$ is sublinear, and then the result follows by the linearity of the Bregman divergence. Define $a_s = x_s^\top\theta$ and $\Delta_s = x_s^\top\lambda$. Then we have $|\Delta_s| = |x_s^\top(\lambda)| \le ||x_s||||\lambda|| \le ||x_s||(||\theta|| + ||\theta + \lambda||) \le (B + B) \le 2B$. Similarly we have $|a_s| \le B$. Hence we satisfy the premise of Lemma 5 and we deduce that

$$
\begin{aligned}
D_{A_s}(\theta + \gamma\lambda, \theta) &= w_s A(x_s^\top\theta + \gamma x_s^\top\lambda) - w_s A(x_s^\top\theta) + \langle x_s w_s A'(x_s^\top\theta), \gamma\lambda \rangle \\
&= w_s(A(a_s + \gamma\Delta_s) - A(a_s) + A'(a_s)\gamma\Delta_s) \\
&\le \frac{w_s}{2}\gamma\left[A(a_s + \gamma\Delta_s) - A(a_s) + A'(a_s)\gamma\Delta_s\right] \\
&= \frac{1}{2}\gamma\left[w_s A(x_s^\top\theta + x_s^\top\lambda) - w_s A(x_s^\top\theta) + w_s A'(x_s^\top\theta)x_s^\top\lambda\right] \\
&= \frac{1}{2}\gamma D_{A_s}(\theta + \lambda, \theta).
\end{aligned}
$$

We also note that for $\gamma \le \frac{\mu}{2L} \le \frac{1}{2}$,

$$D_{A_0}(\theta + \gamma\lambda, \theta) = \frac{\nu}{2}||\gamma\lambda||^2 = \frac{\gamma^2\nu}{2}||\lambda||^2 \le \frac{\gamma\nu}{4}||\lambda||^2 = \frac{1}{2}\gamma D_{A_0}(\theta + \lambda, \theta). \tag{25}$$

Therefore, by summing up the terms, we obtain

$$\mathcal{B}_t(\gamma\lambda) \le \frac{1}{2}\gamma\mathcal{B}_t(\lambda).$$

$\square$

Then it remains to prove that Assumption 1 implies Lemma 5.

*Proof.* (Lemma 5) $L$-Lipschitzness of $A'$ implies smoothness of $A$. Additionally, $\mu$'s existence implies strong convexity of $A$. With this, we can write for any $a$ and $\Delta$ with $a + \Delta \in [-B, B]$

$$A(a + \Delta) \ge A(a) + A'(a)\Delta + \frac{\mu}{2}\Delta^2$$
$$\implies A(a + \Delta) - A(a) - A'(a)\Delta \ge \frac{\mu}{2}\Delta^2.$$

Similarly,

$$A(a + \gamma\Delta) - A(a) - A'(a)\gamma\Delta \le \frac{L}{2}\gamma^2\Delta^2.$$

Putting this together, we have

$$A(a + \gamma\Delta) - A(a) - A'(a)\gamma\Delta \le \frac{L}{2}\gamma^2\Delta^2 = \frac{L\gamma^2}{\mu}\frac{\mu}{2}\Delta^2 \le \frac{L\gamma}{\mu}\gamma[A(a + \Delta) - A(a) - A'(a)\Delta].$$

The question is therefore: when is $\frac{L\gamma}{\mu} \le \frac{1}{2}$? Clearly, choosing $\gamma_0 = \frac{\mu}{2L}$ makes $\frac{L\gamma}{\mu} \le \frac{1}{2}$ for all $\gamma \le \gamma_0$. $\square$

# C FTRL Results: Proofs

## C.1 Technical Lemmas I: Exponential Families

**Lemma 6** (MGF for Exponential family)**.**

$$\mathbb{E}[\exp(T(y)u)|x] = \exp(A({\theta_\star}^\top x + u) - A({\theta_\star}^\top x)).$$

*Proof.*

$$
\begin{aligned}
\mathbb{E}[\exp(T(y)u)|x] &= \int_y \exp(T(y)u)h(y)\exp(T(y){\theta_\star}^\top x - A({\theta_\star}^\top x))dy \\
&= \int \exp(T(y)({\theta_\star}^\top x + u))h(y)\exp(-A({\theta_\star}^\top x)) \\
&\quad\times \exp(-A({\theta_\star}^\top x + u))\exp(A({\theta_\star}^\top x + u))dy \\
&= \exp(A({\theta_\star}^\top x + u) - A({\theta_\star}^\top x)),
\end{aligned}
$$

where the last step follows because the density of a new exponential family distribution with parameter ${\theta_\star}^\top x + u$ also integrates to 1. $\qquad\square$

## C.2 Technical Lemmas II: Elliptical Potential Lemma

We will repeatedly use instantiations of the following key lemma, known as the elliptical potential lemma. We will use the version from Hazan et al. (2006). Other variants are stated in Abbasi-Yadkori et al. (2011) or Carpentier et al. (2020).

**Lemma 7** (Lemma 11 in Hazan et al. (2006))**.** *Let $u_s \in \mathbb{R}^d$ be a sequence of vectors such that $||u_s|| \le r$. Define $\bar{\mathbf{V}}_t = \sum_{s=1}^t u_s u_s^\top + \lambda\mathbf{I}$. Then*

$$\sum_{s=1}^t ||u_s||_{\bar{\mathbf{V}}_s^{-1}}^2 \le \log\left(\frac{\det \bar{\mathbf{V}}_t}{\det \lambda\mathbf{I}}\right) \le d\log\left(\frac{r^2 t}{\lambda} + 1\right).$$

We will also need a result where the time indices of the matrix are shifted. For this, note that if $\lambda \ge r^2$, then $u_s u_s^\top \preceq r^2\mathbf{I} \preceq \lambda\mathbf{I}$, and so we get $\bar{\mathbf{V}}_s \le \bar{\mathbf{V}}_{s-1} + u_s u_s^\top \preceq \bar{\mathbf{V}}_{s-1} + \lambda\mathbf{I} \preceq 2\bar{\mathbf{V}}_{s-1}$. Under our conditions, it follows that

$$\sum_{s=1}^t ||u_s||_{\bar{\mathbf{V}}_{s-1}^{-1}}^2 \le 2\sum_{s=1}^t ||u_s||_{\bar{\mathbf{V}}_s^{-1}}^2$$

**Corollary 1.** *We have the following bounds:*

$$\gamma_t^\lambda = \log\left(\frac{\det(\sum_{s=1} \mu x_s x_s^\top + \lambda\mathbf{I})}{\det(\lambda\mathbf{I})}\right) \le d\log\left(\frac{\mu t}{\lambda} + 1\right),$$

*and*

$$\sum_{s=1}^t ||x_s||_{(\mathbf{V}_{s-1}^{\mu;\lambda})^{-1}}^2 \le \frac{2}{\mu}\gamma_t^\lambda.$$

*Proof.* The first bound is trivial by instantiating $u_s = \sqrt{\mu}x_s$. The second bound is by noting

$$\sum_{s=1}^t ||x_s||_{(\mathbf{V}_{s-1}^{\mu;\lambda})^{-1}}^2 = \frac{1}{\mu}\sum_{s=1}^t ||u_s||_{(\mathbf{V}_{s-1}^{\mu;\lambda})^{-1}}^2 \le \frac{2}{\mu}\sum_{s=1}^t ||u_s||_{(\mathbf{V}_s^{\mu;\lambda})^{-1}}^2 \le \frac{2}{\mu}\gamma_t^\lambda.$$

$\qquad\square$

### C.3 Technical Lemmas III: Supermartingales

**Lemma 8** (Martingale Increment). *Define the parametrized random processes*

$$\mathcal{M}_j(r) = \exp(\nabla f_j(\theta_\star)^\top r - A'(x_j^\top \theta_\star)x_j^\top r - A(x_j^\top \theta_\star - x_j^\top r) + A(x_j^\top \theta_\star))$$

*and*

$$\mathcal{N}_j(r) = \exp(\nabla f_j(\theta_\star)^\top r - \frac{L}{2}r^\top x_j x_j^\top r).$$

*Then, under Assumption 1 we have for any $r \in \mathbb{R}^d$ that $\mathbb{E}[\mathcal{M}_j(r)\,|\,\mathcal{F}_{j-1}] = 1$ and $\mathbb{E}[\mathcal{N}_j(r)\,|\,\mathcal{F}_{j-1}] \leq 1$.*

*Proof.* First, using the form of the exponential family and and recalling that $\nabla_\theta f_j(\theta) = -\nabla_\theta \log p_\theta(y_j\,|\,x_j) = \nabla_\theta[A(x_j^\top \theta) - T(y_j)x_j^\top \theta]$ we obtain

$$\mathbb{E}[\exp(\nabla f_j(\theta_\star)^\top r)\,|\,\mathcal{F}_{j-1}]$$

$$= \int_y \exp(\nabla f_j(\theta_\star)^\top r) \times h(y)\exp(T(y)x_j^\top \theta_\star - A(x_j^\top \theta_\star))dy$$

$$= \int_y \exp(-T(y)x_j^\top r + A'(x_j^\top \theta_\star)x_j^\top r) \times h(y)\exp(T(y)x_j^\top \theta_\star - A(x_j^\top \theta_\star))dy$$

$$= \exp(A'(x_j^\top \theta_\star)x_j^\top r)\underbrace{\int_y h(y)\exp(T(y)(x_j^\top \theta_\star - x_j^\top r))\exp(-A(x_j^\top \theta_\star - x_j^\top r))dy}_{=1}$$

$$\times \exp(A(x_j^\top \theta_\star - x_j^\top r))\exp(-A(x_j^\top \theta_\star))$$

$$= \exp(A'(x_j^\top \theta_\star)x_j^\top r)\exp(A(x_j^\top \theta_\star - x_j^\top r))\exp(-A(x_j^\top \theta_\star)),$$

which finishes the proof. The second statement follows by using $L$-smoothness on the last equation and therefore noting that $\mathcal{N}_j(r) \leq \mathcal{M}_j(r)$. $\qquad\square$

**Lemma 9.** *(Sequential Mixing) Define the martingale process,*

$$M_t(r_1, \ldots r_t) = \prod_{s=1}^t \mathcal{N}_s(r_s),$$

*and recursively define the mixture martingale,*

$$\bar{M}_s = \bar{M}_{s-1} \times \int_r \mathcal{N}_s(r)p_s(r)\mathrm{d}r,$$

*where $p_s$ is a probability distribution equal $\mathcal{N}(0, \mathbf{H}_s^{-1})$, $\mathbf{H}_s = \sum_{j=1}^{s-1} Lx_j x_j^\top + \mathbf{I}\lambda\frac{L}{\mu}$, and $\bar{M}_0 = 1$. Then the following statements hold*

- *$\{\bar{M}_s\}_s$ is an adapted super-martingale with respect to the usual filtration.*

- *$\bar{M}_t = \exp(\frac{\mu}{L}\sum_{s=1}^t \nabla f_s(\theta_\star)^\top(\mathbf{V}_s^{\mu;\lambda})^{-1}\nabla f_s(\theta_\star))\sqrt{\frac{\det(\mathbf{I}\lambda)}{\det(\mathbf{V}_s^{\mu;\lambda})}}.$*

*where*

$$\mathbf{V}_s^{\mu;\lambda} = \sum_{j=1}^s \mu x_j x_j^\top + \lambda\mathbf{I}.$$

*Proof.* The first point follows from the fact that $p_s(r)$ is deterministic conditioned on the sub-$\sigma$-algebra $\mathcal{F}_{s-1}$ (since $p_s$ makes use of $x_s$ but not $x_{s+1}$). Therefore, under mild regularity conditions

$$\mathbb{E}[\bar{M}_s\,|\,\mathcal{F}_{s-1}] = \mathbb{E}\left[\bar{M}_{s-1}\int p_s(r)\mathcal{N}_s(r)dr\,|\,\mathcal{F}_{s-1}\right] = \bar{M}_{s-1}\int_r p_s(r)\,\mathbb{E}[\mathcal{N}_s(r)\,|\,\mathcal{F}_{s-1}]dr \leq \bar{M}_{s-1}.$$

In other words, mixing does not affect the supermartingale properties. For the second point, we derive an explicit form of the mixture martingale. Note that we can write out

$$\int_r \mathcal{N}_s(r) p_s(r) \mathrm{d}r = \frac{1}{\sqrt{(2\pi)^d \det(\mathbf{H}_s^{-1})}} \int_r \exp\left(\nabla f_s(\theta_\star)^\top r - \frac{L}{2}\|r\|_{x_s x_s^\top}^2 - \frac{1}{2}r^\top \mathbf{H}_s r\right)\mathrm{d}r. \tag{26}$$

We can complete the square to obtain

$$\nabla f_s(\theta_\star)^\top r - \frac{L}{2}\|r\|_{x_s x_s^\top}^2 - \frac{1}{2}r^\top \mathbf{H}_s r$$
$$= \frac{1}{2}\|\nabla f_s(\theta_\star)\|_{(\mathbf{H}_s + L x_s x_s^\top)^{-1}}^2 - \frac{1}{2}\|r - (\mathbf{H}_s + L x_s x_s^\top)^{-1}\nabla f_s(\theta_\star)\|_{\mathbf{H}_s + L x_s x_s^\top}^2.$$

The second term is the exponent of a exponent of a Gaussian integral with covariance $\mathbf{H}_{s+1}^{-1}$, and therefore results in

$$\int_r \exp\left(-\frac{1}{2}\|r - (\mathbf{H}_s + L x_s x_s^\top)^{-1}\nabla f_s(\theta_\star)\|_{\mathbf{H}_s + L x_s x_s^\top}^2\right)dr = \sqrt{(2\pi)^d \det(\mathbf{H}_{s+1}^{-1})}.$$

Plugging this into (26) we get

$$\int_r \mathcal{N}_s(r) p_s(r)\mathrm{d}r = \sqrt{\frac{\det \mathbf{H}_s}{\det \mathbf{H}_{s+1}}} \exp\left(\frac{1}{2}\|\nabla f_s(\theta_\star)\|_{\mathbf{H}_{s+1}^{-1}}^2\right).$$

By multiplying the individual steps, we can see that the determinant terms cancel in a telescoping product. This leads to the formulation

$$\bar{M}_t = \exp\left(\sum_{s=1}^t \nabla f_s(\theta_\star)^\top \mathbf{H}_{s+1}^{-1}\nabla f_s(\theta_\star)\right)\sqrt{\frac{\det \frac{\lambda L}{\mu}\mathbf{I}}{\det \mathbf{H}_{t+1}}}.$$

To conclude the proof, note that $\mathbf{H}_{s+1} = \frac{L}{\mu}\mathbf{V}_s^{\mu;\lambda}$. $\qquad\square$

**Lemma 10.** *Under assumption of Lemma 9,*

$$\mathbb{P}\left(\sum_{s=1}^t \|\nabla f_s(\theta_\star)\|_{(\mathbf{V}_s^{\mu;\lambda})^{-1}}^2 \le \frac{L}{\mu}\log\left(\frac{\det(\mathbf{V}_s^{\mu;\lambda})}{\det(\mathbf{I}\lambda)}\right) + \frac{L}{\mu}\log\left(\frac{1}{\delta}\right)\right) \le \delta. \tag{27}$$

*with probability $1 - \delta$.*

*Proof.* The statement, follows by applying Ville's inequality for supermartingales, applying the logarithm, and rearranging. Namely,

$$\mathbb{P}(\bar{M}_t \ge \delta) = \mathbb{P}(\log(\bar{M}_t) \ge \log(\delta)) \le \delta.$$

$\qquad\square$

The following results allow us to upper bound the weighted regret by the unweighted regret:

**Lemma 11** (Weighting Reduction). *Let $\{\theta_s\}_{s=1}^t$ be a sequence of vectors adapted to the filtration $\{\mathcal{F}_{s-1}\}_s$. Define*

$$\Delta_t(\{\theta_s\}) = \sum_{s=1}^t w_s(f_s(\theta_s) - f_s(\theta_\star)) - f_s(\theta_s) + f_s(\theta_\star) = \sum_{s=1}^t (1 - w_s)(f_s(\theta_\star) - f_s(\theta_s)).$$

*Then, $P_t = \exp(\Delta_t(\{\theta_s\}_s))$ is a non-negative super-martingale for any choice of adapted $\{w_s\}$, and hence,*

$$\sum_{s=1}^t w_s(f_s(\theta_s) - f_s(\theta_\star)) \le \sum_{s=1}^t (f_s(\theta_s) - f_s(\theta_\star)) + \log\left(\frac{1}{\delta}\right)$$

*with probability $1 - \delta$ for all $t \ge 0$.*

*Proof.*

$$
\begin{aligned}
\mathbb{E}[P_t \,|\, \mathcal{F}_{t-1}] &= \mathbb{E}_{\theta_\star}\left[\exp\left(\sum_{s=1}^{t} -(1-w_s)f_s(\theta_s) + (1-w_s)f_s(\theta_\star)\right)\Big|\mathcal{F}_{t-1}\right] \\
&= P_{t-1}\,\mathbb{E}_{y_t \sim \mathbb{P}_\star}\exp(-(1-w_t)f_t(\theta_t) + (1-w_t)f_t(\theta_\star)) \\
&= P_{t-1}\int_{y_t}\exp(-(1-w_t)f_t(\theta_t) + (1-w_t)f_t(\theta_\star))\exp(-f_t(\theta_\star))dy_t \\
&= P_{t-1}\int_{y_t}\exp(-(1-w_t)f_t(\theta_t) - w_t f_t(\theta_\star))dy_t \\
&= P_{t-1}\int_{y_t} p_{\theta_t}(y_t \,|\, x_t)^{1-w_t} p_{\theta_\star}(y_t \,|\, x_t)^{w_t}dy_t \\
&= P_{t-1}\exp(-(1-w_t)D_{w_t}(\theta_\star, \theta_t)) \le P_{t-1}.
\end{aligned}
$$

We have used here the definition of the Renyi-divergence and the fact that it is always non-negative, namely

$$
D_w(\theta_1, \theta_2) = \frac{1}{w-1}\log\int_y p_{\theta_1}(y \,|\, x)^{1-w}p_{\theta_1}(y \,|\, x)^w dy \ge 0,
$$

for $0 < w \ne 1$.[3] The rest follows by the application of Ville's inequality. $\qquad\square$

### C.4  FTRL Proof: the Unweighted Case

*Proof of Theorem 4 (first part).* We define the function that FTRL minimizes in each step (to pick $\hat{\theta}_t$) as, $g_t(\theta) = \sum_{s=1}^{t-1} -\log p_\theta(y_s \,|\, x_s) + \lambda||\theta||_2^2$. We can rewrite this objective as

$$
\hat{\theta}_t = \arg\min_{\theta\in\Theta} g_t(\theta) = \arg\min_{\theta\in\Theta}\sum_{s=1}^{t-1} f_s(\theta) + \lambda||\theta||_2^2 = \arg\min_{\theta\in\Theta}\sum_{s=1}^{t-1} m_s(\theta) + \phi_t(\theta),
$$

where we recall that[4]

$$
f_s(\theta) = A(x_s^\top\theta) - T(y_s)x_s^\top\theta - \log h(y_s),
$$

and we have introduced the shorthands

$$
m_s(\theta) = -T(y_s)x_s^\top\theta
$$

and

$$
\phi_t(\theta) = \sum_{s=1}^{t-1} A(\theta^\top x_s) + \lambda||\theta||_2^2.
$$

In essence, we have shifted some of the objective into what is commonly looked at as the regularizer. By a standard telescoping sum argument, we obtain for any $u$

$$
\begin{aligned}
&\sum_{s=1}^{t}(m_s(\hat{\theta}_s) - m_s(u)) \\
={}& \phi_{t+1}(u) - \min_\theta \phi_1(\theta) + \sum_{s=1}^{t}[g_s(\hat{\theta}_s) - g_{s+1}(\hat{\theta}_{s+1}) + m_s(\hat{\theta}_s)] + \underbrace{g_{t+1}(\hat{\theta}_{t+1}) - g_{t+1}(u)}_{\le 0} \\
\le{}& \phi_{t+1}(u) + \sum_{s=1}^{t}[g_s(\hat{\theta}_s) - g_{s+1}(\hat{\theta}_{s+1}) + m_s(\hat{\theta}_s)] \\
={}& \phi_{t+1}(u) + \sum_{s=1}^{t}[g_s(\hat{\theta}_s) - g_{s+1}(\hat{\theta}_{s+1}) + g_{s+1}(\hat{\theta}_s) - \phi_{s+1}(\hat{\theta}_s) - g_s(\hat{\theta}_s) + \phi_s(\hat{\theta}_s)]
\end{aligned}
$$

---

[3]The case $w_t = 1$ is trivial for us.

[4]The $\log h(y_s)$ term does not play any role in the regret nor the FTRL objective.

$$= \quad \phi_{t+1}(u) + \sum_{s=1}^{t} [g_{s+1}(\hat{\theta}_s) - g_{s+1}(\hat{\theta}_{s+1}) - \phi_{s+1}(\hat{\theta}_s) + \phi_s(\hat{\theta}_s)].$$

Now we use the strong-convexity of $g_{s+1}$ under the norm $||\cdot||_{\mathbf{V}_s^{\mu;\lambda}}$ where $\mathbf{V}_s^{\mu;\lambda} = \sum_{j=1}^{s} \mu x_s x_s^\top + \lambda \mathbf{I}$,

$$\sum_{s=1}^{t} (m_s(\hat{\theta}_s) - m_s(u))$$

$$\leq \quad \phi_{t+1}(u) + \sum_{s=1}^{t} [(\hat{\theta}_s - \hat{\theta}_{s+1})^\top \nabla g_{s+1}(\hat{\theta}_s)$$

$$- \frac{1}{2}(\hat{\theta}_s - \hat{\theta}_{s+1})^\top \mathbf{V}_s^{\mu;\lambda}(\hat{\theta}_s - \hat{\theta}_{s+1}) - \phi_{s+1}(\hat{\theta}_s) + \phi_s(\hat{\theta}_s)]$$

$$\leq \quad \phi_{t+1}(u) + \sum_{s=1}^{t} [(\hat{\theta}_s - \hat{\theta}_{s+1})^\top \nabla f_s(\hat{\theta}_s)$$

$$- \frac{1}{2}(\hat{\theta}_s - \hat{\theta}_{s+1})^\top \mathbf{V}_s^{\mu;\lambda}(\hat{\theta}_s - \hat{\theta}_{s+1}) - \phi_{s+1}(\hat{\theta}_s) + \phi_s(\hat{\theta}_s)]$$

$$\leq \quad \phi_{t+1}(u) + \sum_{s=1}^{t} \left[ \frac{1}{2}||\nabla f_s(\hat{\theta}_s)||^2_{(\mathbf{V}_s^{\mu;\lambda})^{-1}} - \phi_{s+1}(\hat{\theta}_s) + \phi_s(\hat{\theta}_s) \right],$$

where in the second inequality we used that $\nabla g_s(\hat{\theta}_s)^\top (x - \hat{\theta}_s) \geq 0$ due to the first-order optimality conditions for convex constrained minimization. Lastly, we optimized the resulting quadratic function over $\hat{\theta}_{s+1}$ (over $\mathbb{R}^d$) to get a worst case bound involving the dual-norm.

Note that for the shorthands we defined above:

$$\sum_{s=1}^{t} [-\phi_{s+1}(\hat{\theta}_s) + \phi_s(\hat{\theta}_s)] = \sum_{s=1}^{t} -A(\hat{\theta}_s^\top x_s).$$

Using our previous observations and the definition of $\phi_{t+1}(\theta_\star)$, we get for the overall regret:

$$\mathcal{R}_t$$

$$= \sum_{s=1}^{t} f_s(\hat{\theta}_s) - f_s(\theta_\star)$$

$$= \sum_{s=1}^{t} m_s(\hat{\theta}_s) - m_s(\theta_\star) + \sum_{s=1}^{t} A(x_s^\top \hat{\theta}_s) - A(x_s^\top \theta_\star)$$

$$= \sum_{s=1}^{t} A(x_s^\top \theta_\star) - A(x_s^\top \hat{\theta}_s) + \sum_{s=1}^{t} A(x_s^\top \hat{\theta}_s) - A(x_s^\top \theta_\star) + \frac{1}{2} \sum_{s=1}^{t} ||\nabla f_s(\hat{\theta}_s)||^2_{(\mathbf{V}_s^{\mu;\lambda})^{-1}} + \lambda ||\theta_\star||^2$$

$$\leq \frac{1}{2} \sum_{s=1}^{t} ||\nabla f_s(\hat{\theta}_s)||^2_{(\mathbf{V}_s^{\mu;\lambda})^{-1}} + \lambda ||\theta_\star||^2$$

$$\leq \frac{1}{2} \sum_{s=1}^{t} ||T(y_s)x_s - A'(x_s^\top \hat{\theta}_s)x_s||^2_{(\mathbf{V}_s^{\mu;\lambda})^{-1}} + \lambda ||\theta_\star||^2$$

$$\leq \sum_{s=1}^{t} \left[ ||T(y_s)x_s - A'(x_s^\top \theta_\star)x_s||^2_{(\mathbf{V}_s^{\mu;\lambda})^{-1}} + ||(A'(x_s^\top \hat{\theta}_s) - A'(x_s^\top \theta_\star))x_s||^2_{(\mathbf{V}_s^{\mu;\lambda})^{-1}} \right] + \lambda ||\theta_\star||^2$$

$$\leq \sum_{s=1}^{t} \left[ ||T(y_s)x_s - A'(x_s^\top \theta_\star)x_s||^2_{(\mathbf{V}_s^{\mu;\lambda})^{-1}} + 2L^2 B^2 ||x_s||^2_{(\mathbf{V}_s^{\mu;\lambda})^{-1}} \right] + \lambda ||\theta_\star||^2$$

$$\leq \sum_{s=1}^{t} \left[ ||\nabla f_s(\theta_\star)||^2_{(\mathbf{V}_s^{\mu;\lambda})^{-1}} + 2L^2 B^2 ||x_s||^2_{(\mathbf{V}_s^{\mu;\lambda})^{-1}} \right] + \lambda B^2$$

$$\leq \lambda B^2 + \frac{L}{\mu}\left(\gamma_t^\lambda + \log\left(\frac{1}{\delta}\right)\right) + \sum_{s=1}^{t} 2L^2 B^2 ||x_s||^2_{(\mathbf{V}_s^{\mu;\lambda})^{-1}}$$

$$\leq \lambda B^2 + \frac{L}{\mu}\left(\gamma_t^\lambda + \log\left(\frac{1}{\delta}\right)\right) + \frac{2L^2 B^2}{\mu}\gamma_t^\lambda.$$

The last line follows because of Lemma 7, and the second to last one follows because of Lemma 10. Notice that if we wish to deal with arbitrary weights $\{w_t\}$, we can simply resort to Lemma 11 and bound the weighted case with the unweighted case. In that case, we incur an additional additive $\log(1\delta)$ term. $\qquad\square$

## C.5 FTRL Analysis: the Weighted Case (Vovk-Azoury-Warmuth Forecaster)

*Proof.* We define the function that FTRL minimizes in each step (to pick $\hat{\theta}_t$) as $\tilde{g}_t(\theta) = \sum_{s=1}^{t-1}[A(x_s^\top \theta) - T(y_s)x_s^\top \theta] + \psi_t(\theta)$ with $\psi_t(\theta) = A(x_t^\top \theta) + \lambda||\theta||_2^2$. We can rewrite this objective as

$$\hat{\theta}_t = \arg\min_{\theta\in\Theta} \tilde{g}_t(\theta) = \arg\min_{\theta\in\Theta} \sum_{s=1}^{t-1} m_s(\theta) + \phi_t(\theta),$$

by introducing the shorthands

$$m_s(\theta) = -T(y_s)x_s^\top \theta$$

and (notice the difference in time index of the second sum when compared to the proof in the previous subsection):

$$\phi_t(\theta) = \sum_{s=1}^{t} A(\theta^\top x_s) + \lambda||\theta||_2^2.$$

In addition consider the objective $g_t$ from the classical FTRL analysis in Section C.4. It is not used to run the online algorithm, but is helpful in our analysis. With our new components, it is equal to

$$g_t(\theta) = \sum_{s=1}^{t-1} m_s(\theta) + \sum_{s=1}^{t-1} A(\theta^\top x_s) + \lambda||\theta||_2^2 = \sum_{s=1}^{t-1} m_s(\theta) + \phi_{t-1}(\theta),$$

and its minimizer is $\bar{\theta}_t = \arg\min_{\theta\in\Theta} g_t(\theta)$. Also, consider a weighted version of the regularizer

$$\bar{\phi}_t(\theta) = \sum_{s=1}^{t} w_s A(x_s^\top \theta) + \lambda||\theta||_2^2,$$

which will be useful. We use a variant of a similar telescoping sum argument as in the previous proof of Section C.4. We specifically use $\theta_\star$ as the comparator to compete against. Notice that we insert a telescoping sum involving the objective $g_s$, which is not the objective that our estimator is minimizing:

$$\sum_{s=1}^{t} w_s(m_s(\hat{\theta}_s) - m_s(\theta_\star))$$

$$\overset{(*)}{=} \bar{\phi}_t(\theta_\star) - \phi_0(\bar{\theta}_1) + \sum_{s=1}^{t}[g_s(\bar{\theta}_s) - g_{s+1}(\bar{\theta}_{s+1}) + w_s m_s(\hat{\theta}_s)] + \underbrace{g_{t+1}(\bar{\theta}_{t+1}) - g_{t+1}(\theta_\star)}_{\leq 0}$$

$$+ \sum_{s=1}^{t}(1 - w_s)f_s(\theta_\star)$$

$$\leq \bar{\phi}_t(\theta_\star) + \sum_{s=1}^{t}[w_s(g_s(\bar{\theta}_s) - g_{s+1}(\bar{\theta}_{s+1})) + w_s m_s(\hat{\theta}_s)]$$

$$+ \sum_{s=1}^{t}(1 - w_s)(g_s(\bar{\theta}_s) - g_{s+1}(\bar{\theta}_{s+1}) + f_s(\theta_\star))$$

$$\overset{(**)}{=} \quad \bar{\phi}_t(\theta_\star) + \sum_{s=1}^{t} w_s[g_s(\bar{\theta}_s) - g_{s+1}(\bar{\theta}_{s+1}) + g_{s+1}(\hat{\theta}_s) - \phi_s(\hat{\theta}_s) - g_s(\hat{\theta}_s) + \phi_{s-1}(\hat{\theta}_s)]$$

$$\sum_{s=1}^{t}(1 - w_s)[g_s(\bar{\theta}_s) - g_{s+1}(\bar{\theta}_{s+1}) + f_s(\theta_\star)]$$

$$\overset{(***)}{\leq} \quad \bar{\phi}_t(\theta_\star) + \sum_{s=1}^{t} w_s[-g_{s+1}(\bar{\theta}_{s+1}) + g_{s+1}(\hat{\theta}_s) - \phi_s(\hat{\theta}_s) + \phi_{s-1}(\hat{\theta}_s)] + \tilde{\Delta}_t.$$

In $(*)$, we used the shorthands and definitions introduced above. In $(**)$, we used the identity $g_{s+1}(\hat{\theta}_s) - \phi_s(\hat{\theta}_s) - g_s(\hat{\theta}_s) + \phi_{s-1}(\hat{\theta}_s) = m_s(\hat{\theta}_s)$. Finally, for $(***)$, recall that $\bar{\theta}_s$ is the minimizer of $g_s$, and hence, $g_s(\bar{\theta}_s) - g_s(\hat{\theta}_s) \leq 0$. Next, define $\tilde{\Delta}_t = \sum_{s=1}^{t}(1 - w_s)[g_s(\bar{\theta}_s) - g_{s+1}(\bar{\theta}_{s+1}) + f_s(\theta_\star)]$. We will bound this term later.

Now, we use the strong-convexity of $g_{s+1}(\theta)$ under the norm $||\cdot||_{\mathbf{V}_s^{\mu;\lambda}}$ where $\mathbf{V}_s^{\mu;\lambda} = \sum_{j=1}^{s} \mu x_j x_j^\top + \lambda \mathbf{I}$, namely

$$g_{s+1}(\bar{\theta}_{s+1}) \geq g_{s+1}(\hat{\theta}_{s+1}) + \nabla g_{s+1}(\hat{\theta}_s)^\top (\bar{\theta}_{s+1} - \hat{\theta}_s) + \frac{1}{2}||\bar{\theta}_{s+1} - \hat{\theta}_s||^2_{\mathbf{V}_s^{\mu;\lambda}}.$$

We can then proceed as follows:

$$\sum_{s=1}^{t} w_s(m_s(\hat{\theta}_s) - m_s(u))$$

$$\leq \quad \tilde{\Delta}_t + \bar{\phi}_t(\theta_\star) + \sum_{s=1}^{t} w_s[\nabla g_{s+1}(\hat{\theta}_s)^\top (\hat{\theta}_s - \bar{\theta}_{s+1})$$

$$- \frac{1}{2}||\bar{\theta}_{s+1} - \hat{\theta}_s||^2_{\mathbf{V}_s^{\mu;\lambda}} - \phi_s(\hat{\theta}_s) + \phi_{s-1}(\hat{\theta}_s)]$$

$$\leq \quad \tilde{\Delta}_t + \bar{\phi}_t(\theta_\star) + \sum_{s=1}^{t} w_s[(\nabla \tilde{g}_s(\hat{\theta}_s) + \nabla m_s(\hat{\theta}_s))^\top (\hat{\theta}_s - \bar{\theta}_{s+1})$$

$$- \frac{1}{2}||\bar{\theta}_{s+1} - \hat{\theta}_s||^2_{\mathbf{V}_s^{\mu;\lambda}} - \phi_s(\hat{\theta}_s) + \phi_{s-1}(\hat{\theta}_s)]$$

$$\leq \quad \tilde{\Delta}_t + \bar{\phi}_t(\theta_\star) + \frac{1}{2}\sum_{s=1}^{t} w_s||\nabla m_s(\hat{\theta}_s)||^2_{(\mathbf{V}_s^\mu)^{-1}} - w_s(\phi_s(\hat{\theta}_s) + \phi_{s-1}(\hat{\theta}_s)), \quad (28)$$

where in the second to last line we used that $\nabla \tilde{g}_s(\hat{\theta}_s)^\top (x - \hat{\theta}_s) \geq 0$ for any $x$, due to the optimality of $\hat{\theta}_s$ for the FTRL objective. In the last line, we optimized over $\bar{\theta}_{s+1}$ to get a worst-case bound on the quadratic function involving it. Also, note that for the shorthands we defined above:

$$\sum_{s=1}^{t} w_s(-\phi_s(\hat{\theta}_s) + \phi_{s-1}(\hat{\theta}_s)) = \sum_{s=1}^{t} w_s(-A(\hat{\theta}_s^\top x_s)).$$

Our goal here is to upper bound the overall regret:

$$\mathcal{R}_t \quad = \quad \sum_{s=1}^{t} w_s(f_s(\hat{\theta}_s) - f_s(\theta_\star))$$

$$= \quad \sum_{s=1}^{t} w_s(m_s(\hat{\theta}_s) - m_s(\theta_\star)) + \sum_{s=1}^{t} w_s(A(x_s^\top \hat{\theta}_s) - A(x_s^\top \theta_\star))$$

$$\leq \quad \bar{\phi}_t(\theta_\star) - \sum_{s=1}^{t} w_s A(x_s^\top \hat{\theta}_s) + \frac{1}{2}\sum_{s=1}^{t} w_s||\nabla m_s(\hat{\theta}_s)||^2_{(\mathbf{V}_s^\mu)^{-1}} + \tilde{\Delta}_t$$

$$+ \sum_{s=1}^{t} w_s(A(x_s^\top \hat{\theta}_s) - A(x_s^\top \theta_\star))$$

$$
\begin{aligned}
= \ & \sum_{s=1}^{t} w_s(A(x_s^\top \theta_\star)) - \sum_{s=1}^{t} w_s A(x_s^\top \hat{\theta}_s) + \frac{1}{2}\sum_{s=1}^{t} w_s ||\nabla m_s(\hat{\theta}_s)||^2_{(\mathbf{V}_s^\mu)^{-1}} + \lambda ||\theta_\star||^2 + \tilde{\Delta}_t \\
& + \sum_{s=1}^{t} w_s(A(x_s^\top \hat{\theta}_s) - A(x_s^\top \theta_\star)) \\
= \ & \tilde{\Delta}_t + \frac{1}{2}\sum_{s=1}^{t} w_s ||\nabla m_s(\hat{\theta}_s)||^2_{(\mathbf{V}_s^\mu)^{-1}} + \lambda ||\theta_\star||^2.
\end{aligned}
\tag{29}
$$

The first inequality follows by plugging in (28).

We return back to the term $\tilde{\Delta}_t$,

$$
\begin{aligned}
\tilde{\Delta}_t &= \sum_{s=1}^{t}(1 - w_s)[g_s(\bar{\theta}_s) - g_{s+1}(\bar{\theta}_{s+1}) + f_s(\theta_\star)] \\
&= \sum_{s=1}^{t}(1 - w_s)[g_s(\bar{\theta}_s) - g_s(\bar{\theta}_{s+1}) - f_s(\bar{\theta}_{s+1}) + f_s(\theta_\star)] \\
&\leq \sum_{s=1}^{t}(1 - w_s)(f_s(\theta_\star) - f_s(\bar{\theta}_{s+1})) \\
&\leq \frac{L}{\mu}(\gamma_t^\lambda + \log(1/\delta)),
\end{aligned}
\tag{30}
$$

where the second to last line is by the optimality of $\bar{\theta}_s$ and the last one by the assumption in the theorem.

Carrying on with the analysis, i.e. with (29), we insert the definition of $m_s$ and obtain

$$
\begin{aligned}
& \mathcal{R}_t \\
\leq \ & \Delta_t + \frac{1}{2}\sum_{s=1}^{t} w_s ||-T(y_s)x_s||^2_{(\mathbf{V}_s^{\mu;\lambda})^{-1}} + \lambda ||\theta_\star||^2 \\
\leq \ & \Delta_t + \sum_{s=1}^{t}\left[ w_s ||T(y_s)x_s - A'(x_s^\top \theta_\star)x_s||^2_{(\mathbf{V}_s^{\mu;\lambda})^{-1}} + w_s ||(A'(x_s^\top \theta_\star))x_s||^2_{(\mathbf{V}_s^{\mu;\lambda})^{-1}}\right] + \lambda ||\theta_\star||^2 \\
\overset{(*)}{\leq} \ & \Delta_t + \sum_{s=1}^{t} ||T(y_s)x_s - A'(x_s^\top \theta_\star)x_s||^2_{(\mathbf{V}_s^{\mu;\lambda})^{-1}} + L^2 B^2 \sum_{s=1}^{t} w_s ||x_s||^2_{(\mathbf{V}_s^{\mu;\lambda})^{-1}} + \lambda ||\theta_\star||^2 \\
\overset{(**)}{=} \ & \Delta_t + \sum_{s=1}^{t} ||\nabla f_s(\theta_\star)||^2_{(\mathbf{V}_s^\mu)^{-1}} + L\sum_{s=1}^{t} \frac{B^2}{1/L + \text{bias}^2_{x_s}(\hat{\theta}_s)}||x_s||^2_{(\mathbf{V}_s^{\mu;\lambda})^{-1}} + \lambda B^2 \\
\overset{(***)}{\leq} \ & \lambda B^2 + \frac{2L}{\mu}\left(\gamma_t^\lambda + \log\left(\frac{1}{\delta}\right)\right) + L\sum_{s=1}^{t} \frac{B^2}{1/L + \text{bias}^2_{x_s}(\hat{\theta}_s)}||x_s||^2_{(\mathbf{V}_s^{\mu;\lambda})^{-1}}.
\end{aligned}
$$

In $(*)$, we use $w_s \leq 1$ and the Lipschitzness of $A'$. In $(**)$, we use the definition of the weights. Finally, in $(***)$, we used Lemma 10 and (30). By substituting $\Delta \gamma_s = \mu ||x_s||^2_{(\mathbf{V}_s^{\mu;\lambda})^{-1}}$, we finish the proof. The event in Lemma 10 holds with probability $1 - \delta$, completing the proof. $\qquad \square$

### C.6 FTRL Analysis: Beyond Global Smoothness

In this subsection, we give alternative analysis which avoids the necessity to impose a global smoothness condition our likelihood; instead strong convexity within a bounded domain suffices, and we will only assume that

$$
\epsilon_s := \underset{x_s^\top \theta_\star}{\mathbb{E}}[T(y_s)] - T(y_s)
$$

are sub-Exponential random variables, setting us apart from Zhao et al. (2022) which assume sub-Gaussianity. In particular, we can show the following theorem

**Theorem 7.** *With probability $1 - \delta$, uniformly over time $t \in \mathbb{N}$, we have*

$$\mathcal{R}_t \leq cd \log^2(t/\delta)) \log(t),$$

*where the universal constant $c$ hides all constants independent of $t, d$ and $\delta$.*

### C.6.1  Lemmas

We state the following result on sub-Exponential random variables.

**Proposition 2** (Theorem 2.13 in Wainwright (2019))**.** *If $X$ is a centered sub-Exponential variable with some finite variance proxy, then there exist constants $c_1, c_2 > 0$ such that for any $t > 0$*

$$\mathbb{P}(|X| \geq a) \leq c_1 e^{-c_2 a}.$$

By some careful union bounds (akin to a stitching argument), we can also provide upper bounds on anytime-valid upper bounds on the process $S_t = \max_{s \leq t} \epsilon_s$.

**Lemma 12.** *For any sequence $(\epsilon_s)_{s=1}^{\infty}$ of sub-Exponential-variables, there exists a constant $\tilde{c}$ independent of $t$ such that*

$$\mathbb{P}(\exists t : \max_{s \leq t} |\epsilon_s| \geq \tilde{c} \log(s/\delta)) \leq \delta.$$

*Proof.* (of Lemma 12) By Proposition 2, there exists $c_1, c_2 > 0$ such that we have $\mathbb{P}(|\epsilon_s| \geq a) \leq c_1 e^{-c_2 a}$ for any fixed $s$. Note that $c_1 e^{-c_2 a} \leq \delta$ is satisfied for $a \geq \frac{1}{c_2} \log(c_1/\delta) =: c_3 \log(c_1/\delta)$. Let us denote by $\mathcal{E}_i$ the event all $j \in [2^i, 2^{i+1}) \cap \mathbb{N}$ satisfy the inequality

$$|\epsilon_j| < c_3 \log(c_1(2^{2i+1})/\delta).$$

For a single $j$, this happens with probability at least $1 - \delta/2^{2i+1}$. Therefore, by a union bound, as $|[2^i, 2^{i+1}) \cap \mathbb{N}| = 2^i$, we can bound the probability of the complement, namely $\mathbb{P}(\mathcal{E}_i^c) \leq 2^i \frac{\delta}{2^{2i+1}} = \frac{\delta}{2^{i+1}}$. Now, by another union bound, we can conclude that

$$\mathbb{P}(\cup_{i=0}^{\infty} \mathcal{E}_i^c) \leq \sum_{i=0}^{\infty} \frac{\delta}{2^{i+1}} = \frac{\delta}{2} \frac{1}{1 - \frac{1}{2}} = \delta.$$

Now we also have for any $j$ in this range that $2^{2i+1} \leq 2j^2$, and therefore, if $\mathcal{E}_i$ holds, we have for any $j \in [2^i, 2^{i+1}) \cap \mathbb{N}$:

$$\epsilon_j \leq c_3 \log(c_1(2j^2)/\delta) \leq 2c_3 \log(2c_1 j/\delta) \leq \tilde{c} \log(j/\delta).$$

We can immediately see that this implies

$$\mathbb{P}(\exists t : \max_{s \leq t} |\epsilon_s| \geq \tilde{c} \log(s/\delta)) \leq \delta.$$

as desired. $\qquad\square$

### C.6.2  Proof of Theorem 7

Our proof initially follows the FTRL regret bound proofs in the adversarial setting Hazan (2016); Orabona (2019). It also has overlap with the proof in Zhao et al. (2022). We define the function that FTRL minimizes in each step as (to pick $\hat{\theta}_t$)

$$g_t(\theta) = \sum_{s=1}^{t-1} -\log p_\theta(y_s \mid x_s) + \phi(\theta)$$

for convenience. We initially use the same steps as in Theorem 4 to see that for any $u \in \Theta$

$$\sum_{s=1}^{t} (f_s(\hat{\theta}_s) - f_s(u))$$

$$\leq \phi(u) - \min_{\theta} \phi(\theta) + \sum_{s=1}^{t} [g_s(\hat{\theta}_s) - g_{s+1}(\hat{\theta}_{s+1}) + f_s(\hat{\theta}_s)] + g_{t+1}(\hat{\theta}_{t+1}) - g_{t+1}(u)$$

$$\leq \lambda B^2 + \sum_{s=1}^{t} [g_s(\hat{\theta}_s) - g_{s+1}(\hat{\theta}_{s+1}) + f_s(\hat{\theta}_s)].$$

Similarly to the proof of Theorem 4 in Appendix C.4 we bound these increments by the dual norm of the gradient of the objective.

$$g_s(\hat{\theta}_s) - g_{s+1}(\hat{\theta}_{s+1}) + f_s(\hat{\theta}_s) \leq \frac{||\nabla f_s(\hat{\theta}_s)||^2_{(\mathbf{V}_s^{\mu;\lambda})^{-1}}}{2}. \tag{31}$$

Now we note that

$$\nabla f_s(\theta) = A'(x_s^\top \theta) x_s - T(y_s) x_s.$$

Using properties of the exponential family, we deduce that

$$\nabla f_s(\theta) = \left( \mathop{\mathbb{E}}_{x_s^\top \theta} [T(y_s)] - T(y_s) \right) x_s$$

$$= \left( \mathop{\mathbb{E}}_{x_s^\top \theta} [T(y_s)] - \mathop{\mathbb{E}}_{x_s^\top \theta_\star} [T(y_s)] + \mathop{\mathbb{E}}_{x_s^\top \theta_\star} [T(y_s)] - T(y_s) \right) x_s.$$

From here on out, we proceed more crudely than in our previous analyses, since we are only concerned with asymptotic behavior when $d$ and $t$ are large. Let us define

$$U := \sup_{\theta \in \Theta} | \mathop{\mathbb{E}}_{x_s^\top \theta} [T(y_s)] - \mathop{\mathbb{E}}_{x_s^\top \theta_\star} [T(y_s)]|,$$

which is a model-dependent, deterministic quantity. Let us define the noise variables

$$\epsilon_s := \mathop{\mathbb{E}}_{x_s^\top \theta_\star} [T(y_s)] - T(y_s).$$

We bound

$$||\nabla f_s(\theta)||^2_{(\mathbf{V}_s^{\mu;\lambda})^{-1}} \leq 2(U^2 + \epsilon_s^2)||x_s||^2_{(\mathbf{V}_s^{\mu;\lambda})^{-1}}.$$

Note that the $\epsilon_s$ are centered, independent sub-Exponential variables, and as such are guaranteed to satisfy

$$\mathbb{P}(\exists t : \max_{s \leq t} |\epsilon_s| \geq c_4 \log(s/\delta)) \leq \delta,$$

by Lemma 12. This tells us that conditional on this event, we can upper bound for any $t$

$$\mathcal{R}_t \leq \lambda B^2 + \frac{1}{\mu} (U^2 + \tilde{c}^2 \log^2(t/\delta)) \sum_{s=1}^{t} ||x_s||^2_{(\mathbf{V}_s^{\mu;\lambda})^{-1}}.$$

for some constant $\tilde{c}$ independent of $t$. By Lemma 7, there is thus a constant $c'$ independent of $t$ and $d$ such that

$$\mathcal{R}_t \leq c' d \log^2(t/\delta)) \log(t) = \mathcal{O}(d \log^3(t)),$$

with probability $1 - \delta$ uniformly over $t \in \mathbb{N}$.

# D   Regret Consequences for Stochastic Linear Bandits

As a corollary of our analysis, we provide the regret for stochastic linear bandits that use our confidence sets within the LinUCB algorithm.

*Proof.* We proceed in two parts: first, we instantiate Theorem 3 and then we follow the classical regret analysis for stochastic linear bandits.

**Specializing the Bregman divergence results** By Theorem 3, we know that for any $\nu > 0$, we have that with probability $1 - \delta$,

$$D_{Z_t^\nu}(\theta, \theta_\star) \leq \frac{4L}{\mu} \xi_t + 2 \log\left(\frac{1}{\delta}\right) + 2\mathcal{R}_t, \tag{32}$$

for all $t$, where $\xi_t = \left(\log\left(\frac{1}{\alpha}\right) + \nu B^2 + \Gamma_t^\nu\right)$ and $\mathcal{R}_t$ is the online convex optimization regret. We also recall that

$$Z_t^\nu(\theta) = \sum_{s=1}^t w_s A(x_s^\top \theta) + \frac{\nu}{2} ||\theta||_2^2.$$

In the Gaussian case, where $A(z) = z^2/(2\sigma^2)$. This implies that

$$\nabla Z_t^\nu(\theta) = \sum_{s=1}^t \frac{w_s}{\sigma^2} x_s x_s^\top \theta + \nu\theta = \mathbf{W}_t^{\sigma^{-2};\nu}\theta,$$

where we have defined a weighted version of $\mathbf{V}_t^{\sigma^{-2};\nu}$ as $\mathbf{W}_t^{\sigma^{-2};\nu} = \sum_{s=1}^t \frac{w_s x_s x_s^\top}{\sigma^2} + \nu\mathbf{I}$ and therefore the Bregman divergence is given by $D_{Z_t^\nu}(\theta, \theta_\star) = \frac{1}{2}||\theta - \theta_\star||^2_{\mathbf{W}_t^{\sigma^{-2};\nu}}$. We can also see that the Bregman information gain is given by

$$\Gamma_t^\nu = \log\left(\frac{\int_{\mathbb{R}^d} \exp(-\frac{1}{2}||\theta||_2^2)d\theta}{\int_{\mathbb{R}^d} \exp(-D_{Z_t^\nu}(\theta, \tilde{\theta}_t))d\theta}\right) = \log\left(\frac{\int_{\mathbb{R}^d} \exp(-\frac{1}{2}||\theta||_2)^2 d\theta}{\int_{\mathbb{R}^d} \exp(-\frac{1}{2}||\theta - \theta_\star||^2_{\mathbf{W}_t^{\sigma^{-2};\nu}})d\theta}\right).$$

These Gaussian integrals are straightforward to evaluate. We know that

$$\int_{\mathbb{R}^d} \exp\left(-\frac{1}{2}||\theta||_2^2\right) d\theta = (2\pi)^{d/2}\sqrt{\det((\nu\mathbf{I}_d)^{-1})}.$$

Similarly,

$$\int_{\mathbb{R}^d} \exp\left(-\frac{1}{2}||\theta - \theta_\star||^2_{\mathbf{W}_t^{\sigma^{-2};\nu}}\right) d\theta = (2\pi)^{d/2}\sqrt{\det((\mathbf{W}_t^{\sigma^{-2};\nu})^{-1})}.$$

Then, we can compute

$$\Gamma_t^\nu = \log\left(\frac{\det(\mathbf{W}_t^{\sigma^{-2};\nu})}{\det(\nu\mathbf{I})}\right) = \log\left(\det\left(\sum_{s=1}^t \frac{w_s x_s x_s^\top}{\sigma^2 \nu} + \mathbf{I}\right)\right).$$

In the unweighted case, with which we proceed, we have $\Gamma_t^\nu = \gamma_t^\nu$, that is we recover the classical upper bound on the information gain (Srinivas et al., 2009). To summarize, we have specialized the bound (32) to say that for any $\theta \in \mathcal{C}_t$, we have (since $L = \mu = 1/\sigma^2$)

$$||\theta - \theta_\star||^2_{\mathbf{V}_t^{\sigma^{-2};\nu}} \leq 8\left(\log(1/\alpha) + \nu B^2 + \gamma_t^\nu\right) + 4\log(3/\delta) + 4\mathcal{R}_t.$$

Now, we instantiate the regret of the online learner using Theorem 4. With probability $1 - \delta$, uniformly over $t$, we have

$$\mathcal{R}_t \leq \lambda B^2 + \frac{L}{\mu}\left(\gamma_t^\lambda + \log\left(\frac{1}{\delta}\right)\right) + \frac{2L^2 B^2}{\mu}\gamma_t^\lambda. \tag{33}$$

We get by chosing $\alpha = \delta$, and setting $\nu = \lambda$ that

$$||\theta - \theta_\star||^2_{\mathbf{V}_t^{\sigma^{-2};\lambda}}$$

$$\leq 8\left(\log(1/\delta) + \nu B^2 + \gamma_t^\nu\right) + 4\log(1/\delta)) + 4\lambda B^2 + 4(\gamma_t^\lambda + \log(1/\delta)) + \frac{8B^2}{\sigma^2}\gamma_t^\lambda$$

$$\leq 16\log(1/\delta) + 12\lambda B^2 + 8\left(\frac{B^2}{\sigma^2} + 1\right)\gamma_t^\lambda =: \beta_t.$$

**Linear bandit regret analysis** We are ready to proceed with the bandit analysis for the UCB Algorithm. We follow Lattimore and Szepesvári (2020) and bound the pseudo-regret, letting $x_\star$ be the optimal action. We bound the instantaneous regret at step $0 \leq s \leq t$ as

$$
\begin{aligned}
r_s &= \langle \theta_\star, \, x_\star - x_s \rangle \\
&= \langle \theta_\star, \, x_\star \rangle - \langle \theta_\star, \, x_s \rangle \\
&\leq \max_{\theta \in \mathcal{C}_{s-1}} \langle \theta, \, x_\star \rangle - \langle \theta_\star, \, x_s \rangle \\
&\leq \max_{x \in \mathcal{X}} \max_{\theta \in \mathcal{C}_{s-1}} \langle \theta, \, x \rangle - \langle \theta_\star, \, x_s \rangle \\
&\overset{(*)}{=} \max_{\theta \in \mathcal{C}_{s-1}} \langle \theta, \, x_s \rangle - \langle \theta_\star, \, x_s \rangle \\
&\overset{(**)}{\leq} ||\tilde{\theta}_s - \theta_\star||_{\mathbf{V}_{s-1}^{\sigma^{-2};\lambda}} ||x_s||_{(\mathbf{V}_{s-1}^{\sigma^{-2};\lambda})^{-1}} \\
&\leq \sqrt{\beta_{s-1}} ||x_s||_{(\mathbf{V}_{s-1}^{\sigma^{-2};\lambda})^{-1}} \\
&\leq \sqrt{\beta_t} ||x_s||_{(\mathbf{V}_{s-1}^{\sigma^{-2};\lambda})^{-1}}.
\end{aligned}
$$

The first inequality replaces $\theta_\star$ by the upper confidence bound for action $x_\star$, which is valid with probability $1 - \alpha = 1 - \delta$ uniformly over time. Then, $(*)$ uses the fact that $x_t$ is chosen to maximize the upper confidence bound. Finally $(**)$ defines the UCB parameter $\tilde{\theta}_t$. By Corollary 1, we have

$$
\sum_{s=1}^{t} ||x_s||^2_{(\mathbf{V}_{s-1}^{\sigma^{-2};\lambda})^{-1}} \leq 2\sigma^2 \gamma_t^\lambda.
$$

Plugging all this together and using an $\ell_1/\ell_2$-norm inequality, we get

$$
\begin{aligned}
\mathfrak{R}_t &= \sum_{s=1}^{t} r_s \\
&\leq \sqrt{\beta_t} \sqrt{t \sum_{s=1}^{t} ||x_s||^2_{(\mathbf{V}_{s-1}^{\sigma^{-2};\lambda})^{-1}}} \\
&\leq \sqrt{2t\beta_t \sigma^2 \gamma_t^\lambda} \\
&\leq \sqrt{2\sigma^2 t (16 \log(1/\delta) + 12\lambda B^2 + 8\gamma_t^\lambda + 8B^2/\sigma^2 \gamma_t^\lambda)\gamma_t^\lambda} \\
&\leq 6\sqrt{t\gamma_t} \left( \sigma \sqrt{\log(1/\delta) + \gamma_t^\lambda} + \sigma \lambda^{1/2} B + B\sqrt{\gamma_t^\lambda} \right).
\end{aligned}
$$

To summarize and to justify why this bound holds with probability $1 - 3\delta$ uniformly over time, note that we have bounded the probability of the FTRL bound (33) not holding for some $t$ by $\delta$. Then, the probability of (32) not holding for some $t$ is at most $\delta$. Finally, the anytime Type I error of our sets is also bounded by $\delta$. A union bound therefore concludes the proof. □

### D.1 Comparison to Abbasi-Yadkori et. al. (2011)

We compare our result to the one from Abbasi-Yadkori et al. (2011). Under the assumption that $\lambda \geq 1$, they show that the regret satisfies

$$
\mathfrak{R}_t \leq 4\sqrt{td \log(\lambda + t/d)} \left( \sqrt{\lambda} B + \sigma \sqrt{2 \log(1/\delta) + d \log(1 + t/(\lambda d))} \right).
$$

Observe that there is a reparametrization for the regularizer to get even more similar bounds. If we take $\lambda = \tilde{\lambda}/\sigma^2$ for some $\tilde{\lambda} \geq 1$, our bound reads as

$$
6\sqrt{t\gamma_t^{\tilde{\lambda}/\sigma^2}} \left( \sigma \sqrt{\log(1/\delta) + \gamma_t^{\tilde{\lambda}/\sigma^2}} + \tilde{\lambda}^{1/2} B + B\sqrt{\gamma_t^{\tilde{\lambda}/\sigma^2}} \right).
$$

Given that by Corollary 1, we have

$$\gamma_t^{\tilde{\lambda}/\sigma^2} \le d \log\left(\frac{t}{\tilde{\lambda}} + 1\right),$$

we get almost matching bounds, up to an additional $B\sqrt{\gamma_t^{\tilde{\lambda}/\sigma^2}}$ term blowing up the regret, which we attribute to the accumulation of bias without the reweighting scheme. The remaining differences are down to using slightly different versions of the elliptical potential lemma, trading off generality and tightness (Abbasi-Yadkori et al., 2011; Hazan et al., 2006; Lattimore and Szepesvári, 2020).

# E  Experimental Details

## E.1  Calibration Plots

In Figure 3 we report the calibration of heuristics as well as other theoretically motivated works. The other theoretically motivated works are very pessimistic and are not appropriately calibrated. Note that one caveat of reporting calibration is that it is very much influenced by the data collection scheme in the sequential regime. In our case we use a bandit algorithm to collect the data. Arguably, in this setting, regret might be a better measure rather than looking at the calibration of the confidence sets. Additionally, the calibration depends on the true value $\theta_\star$. We report the results for zero parameter and a random parameter from a unit ball. We also report results for i.i.d. data.

## E.2  Baselines and Details

In the following section, we describe the baseline we used in the comparison. The details of parameters used in the experiments can be found in Table 2. As there is no explicit statement for sub-exponential variables, we give a formal derivation in Appendix F.

### E.2.1  Sub-Exponential Random Variables: Confidence Sets

For this baseline, we will assume a linear model with additive sub-exponential noise, namely that there is $\theta_* \in \Theta$ such that $y_t = \langle \theta_*, x_t \rangle + \eta_t$, where $\eta_t$ is $(\nu, \gamma)$-conditionally sub-exponential (Wainwright, 2019). We let as usual

$$\mathbf{V}_t^{\nu^{-2};\lambda} = \sum_{s=1}^{t} \frac{x_s x_s^\top}{\nu^2} \lambda \mathbf{I}.$$

With this, one can prove the following time-uniform concentration result:

**Proposition 3.** *For any $k \in (0,1)$, the following holds:*

$$\mathbb{P}\left(\exists t \,:\, ||\hat{\theta}_t - \theta^*||_{\mathbf{V}_t^{\nu^{-2};\lambda}} \ge \sqrt{\beta_{SE}}\right) \le \delta,$$

*where*

$$\sqrt{\beta_{SE}} = \sqrt{\lambda}||\theta_\star||_2 + \sqrt{\lambda}kB + \frac{d}{\sqrt{\lambda}kB}\log\left(\frac{1}{1-k}\right) + \frac{1}{\sqrt{\lambda}kB}\log\left(\frac{(\det(\mathbf{V}_t^{\nu^{-2};\lambda}))^{1/2}}{\delta\det(\sqrt{\lambda}\mathbf{I})}\right).$$

The proof is very similar to prior work Faury et al. (2020); Mutný and Krause (2021), and can be found in Appendix F. This can readily be applied in the survival analysis (after a suitable transformation explained in the main paper) and Laplace noise experiments.

### E.2.2  Poisson Heuristics

We implement two heuristics. One is a Bayesian formulation due to the Laplace method, and one is due to considerations in Mutný and Krause (2021) of how to construct a valid confidence set using the Fisher information. The Laplace method creates an ellipsoidal confidence set using the second-order information evaluated at the penalized maximum likelihood estimator. Namely, the second derivative of the likelihood evaluated at the maximum penalized likelihood $\hat{\theta}_t$ is

$$\mathbf{V}_{\text{laplace}} = \sum_{s=1}^{t} \exp(\hat{\theta}_t^\top x_s) x_s x_s^\top.$$

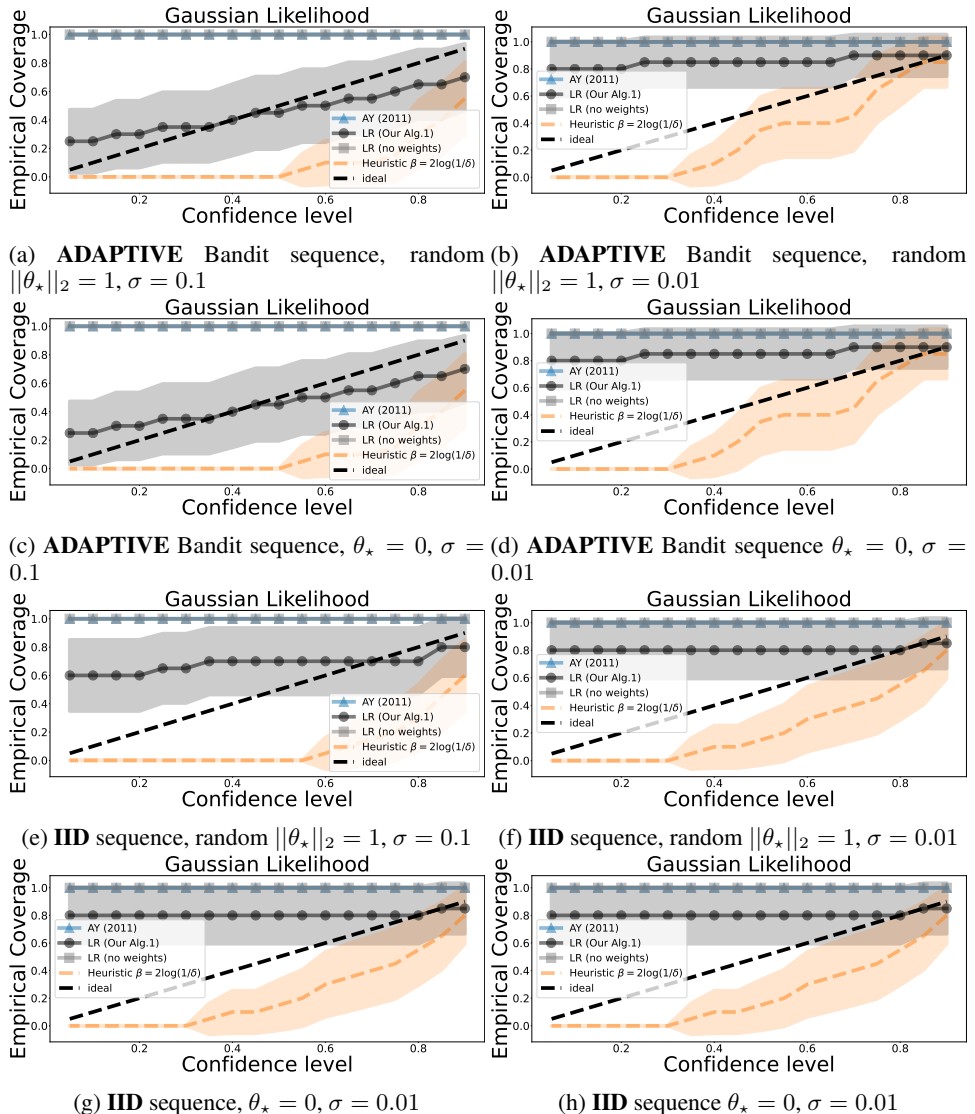

Figure 3: We plot the calibration diagram for data collected from a bandit game trying to optimize a ground truth function using the same model as in Fig. 2a). Instead of the test function, we use an explicit member of the confidence set to avoid a potential mismatch between models. We check after $T = 15$ whether $\theta_\star \in C_t$ and average over 200 runs. We see that the (LR) are more conservative than the ideal calibration, however, they are provably valid and substantially better than any theoretically motivated confidence sets. We also see that the heuristic is not calibrated and fails many times. We see that for i.i.d. data, our sets are somewhat conservative since the data is not adapted, and our approach is not necessary. We note that the results depend on $\theta_\star$.

We use this to define ellipsoidal confidence sets as $||\hat{\theta}_t - \theta||^2_{\mathbf{V}_{\text{laplace}}} \leq 2\log(1/\delta)$. The other heuristic suggests using the worst-case parameter instead, namely

$$\mathbf{V}_{\text{mutny}} = \sum_{s=1}^{t} \exp(B) x_s x_s^\top .$$

This method would have provable coverage with a proper confidence parameter. Its derivation is beyond the scope of this paper.

| Benchmark function | dim | $|\mathcal{X}|$ | $\gamma$ | $B$ | $\lambda$ | Gaussian/Laplace $\sigma/b$ |
|---|---|---|---|---|---|---|
| 1D | 1 | $2^6$ | 0.06 | 4 | 1 | 0.15 |
| Camelback | 2 | $10^2$ | 0.2 | 2 | 1 | 0.10 |

Table 2: Summary of experimental parameters

### E.2.3   NR (2021)

This method follows from Neiswanger and Ramdas (2021). Per se, this method was not developed to be versatile in terms of likelihood but instead provides a confidence set on $f$, even if it originates from a misspecified prior. Nevertheless, it provides a likelihood-aware confidence sequence that is anytime-valid and doesn't employ worst-case parameters, and hence is a good benchmark for our analysis. The confidence sets are of the form

$$\{\theta \mid \log \mathcal{L}(\theta) \leq \log(1/\delta) + \log(p(\mathcal{D}))\},$$

where $\log(p(\mathcal{D}))$ is the current log-evidence of the data given the Gaussian prior. For more information, see Neiswanger and Ramdas (2021).

### E.3   Additive Models

We implemented two likelihoods, namely Gaussian and Laplace. We implemented the discretization of the domain $|\mathcal{X}|$, and in the implementation we used Nystrom features defined on $|\mathcal{X}|$ providing the exact representation of the RKHS on the $\mathcal{X}$, The points were chosen to be on a uniform grid. Notice that for the regularized estimator, we chose the rule of thumb $\lambda = 1/B$ as is motivated in (Mutný and Krause, 2022).

The laplace parameter was picked as $b = 0.15$ likewise. Note that Laplace distribution is sub-exponential with parameters $(b, 2b^2)$. We use $1/\sigma^2$ or $1/b$ respectively for the value $L$. Strictly speaking, the Laplace likelihood is not smooth, but a smoothed objective would most likely inherit a value depending on $b$. As we maintain coverage with any choice of weighting, we do not risk invalidity by using a heuristic choice for $L$.

### E.4   Survival Analysis

We implemented the method exactly as specified, using the Weibull likelihood with parameter $p = 2$. Upon log-transformation, the Gumbel distribution is sub-exponential. To determine the parameter, consider the moment-generating function of the Gumbel distribution ($\beta = 1/p$ in the canonical parameterization):

$$\mathbb{E}[e^X t] = \Gamma(1 - t/2) \exp(t) \leq \exp(t^2/2) \quad \text{for} \quad t < 1/2,$$

hence, the sub-exponentiality parameter is 1, and we can use the above sub-exponential confidence sets with value $b = 1$. For the likelihood ratio code, we used $L = \exp(B)$, as this is the leading term of the Hessian of log-likelihood. The function is not smooth everywhere, but on a bounded domain, this turns out to be an appropriate scaling.

### E.5   Poisson Bandits

In this case, we implemented a bandit game, where we used the parametrization $r_\theta(x) \sim$ Poisson$(\exp(-\theta^\top \Phi(x)))$, where $\Phi(x)$ is the RKHS evaluation operator, and $\theta$ is the unknown value.

We used $L = \exp(B)$, as this is the leading term of the Hessian of log-likelihood in this parametrization. The function is not smooth everywhere but on a bounded domain this is an appropriate scaling.

### E.6   Additional Benchmark Functions

We focus on an additional baseline function: Camelback in 2D, a standard BO benchmark function. The results can be see in Figure 4.

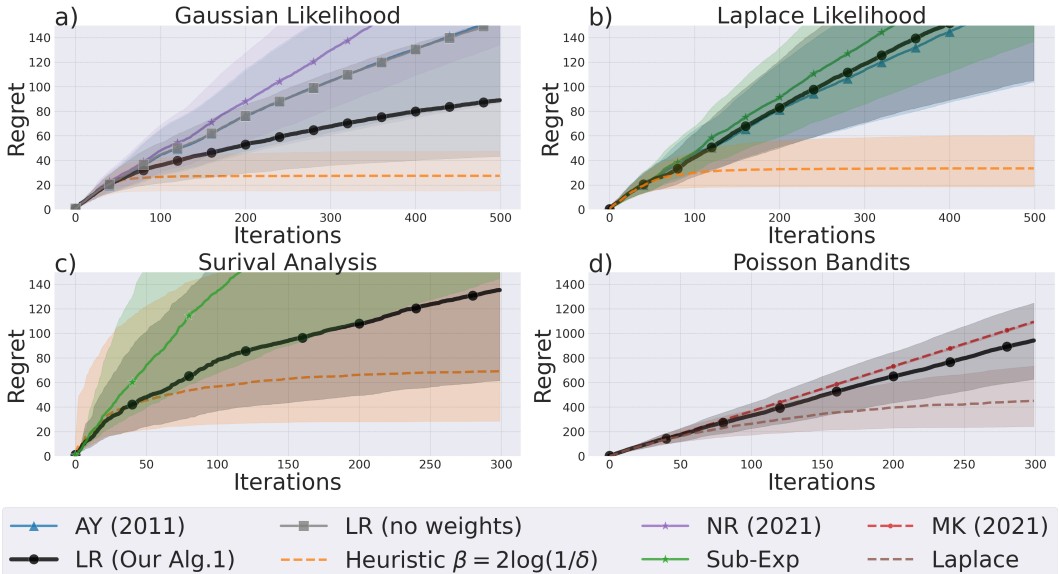

Figure 4: Camelback function. 10 repeats with median and standard quantiles plotted. Note that our method is the best method with provable coverage.

# F    Proof of Proposition 3

Define $S_t$ and the shorthand $\mathbf{V}_t$

$$S_t = \sum_{s=1}^{t} \eta_s \frac{x_s}{\nu^2} \quad \text{and} \quad \mathbf{V}_t := \mathbf{V}_t^{\nu^{-2};\lambda} = \sum_{s=1}^{t} \frac{x_s x_s^\top}{\nu^2} + \lambda \mathbf{I}.$$

and the parametrized process

$$\mathcal{M}_t(x) = \exp(\langle x,\, S_t\rangle - \frac{1}{2}||x||^2_{\mathbf{V}_t}).$$

**Lemma 13.** *If $\eta_t$ is $(\nu, \gamma)$-conditionally sub-Exponential, then $\mathcal{M}_t(x)$ is a super-martingale on the ball $\{x \in \mathbb{R}^d \,|\, ||x||_2 \le \frac{\nu^2}{\gamma}\}$ with $\mathcal{M}_0(x) \le 1$.*

Note that for $\gamma \to 0$ (sub-Gaussian case), this recovers Lemma 20.2. in Lattimore and Szepesvári (2020).

*Proof.* It is easy to observe that for any $x$, we have

$$\exp\left(S_0^\top x - \frac{1}{2}||x||^2_{\mathbf{V}_0}\right) = \exp\left(-\frac{1}{2}||x||^2_{\lambda\mathbf{I}}\right) \le 1.$$

For the first part, we can write

$$\mathbb{E}[\mathcal{M}_t(x)|\mathcal{F}_{t-1}] = \mathbb{E}[\exp(\langle x,\, S_t\rangle - \frac{1}{2}||x||^2_{\mathbf{V}_t})\,|\,\mathcal{F}_{t-1}]$$

$$= \mathbb{E}[\exp(\langle x,\, S_{t-1}\rangle - \frac{1}{2}||x||^2_{\mathbf{V}_{t-1}})\exp(\frac{1}{\nu^2}\langle x,\, \eta_t x_t\rangle - \frac{1}{2\nu^2}||x||^2_{x_t x_t^\top})\,|\,\mathcal{F}_{t-1}]$$

$$= \mathcal{M}_{t-1}(x)\,\mathbb{E}[\exp(\frac{1}{\nu^2}\eta_t\langle x,\, x_t\rangle)\,|\,\mathcal{F}_{t-1}]\exp(-\frac{1}{2\nu^2}||x||^2_{x_t x_t^\top}),$$

where in the last step we use that $x_t$ is $\mathcal{F}_{t-1}$-measurable. Now, as long as $\frac{1}{\nu^2}|\langle x,\, x_s\rangle| \le \frac{1}{\gamma}$, we can apply our definition of conditional sub-Exponential noise to bound

$$\mathbb{E}[\exp(\eta_t\frac{1}{\nu^2}\langle x,\, x_t\rangle)] \le \exp\left(\frac{(\langle x,\, x_t\rangle)^2\nu^2}{2\nu^4}\right) \le \exp\left(\frac{||x||^2_{x_t x_t^\top}}{2\nu^2}\right).$$

From this we directly conclude

$$\mathbb{E}[\mathcal{M}_t(x)|\mathcal{F}_{t-1}] \leq \mathcal{M}_{t-1}(x).$$

By Cauchy-Schwarz, a sufficient condition is $||x||_2 \leq \frac{\nu^2}{\gamma}$ as this implies (with our assumptions on the actions)

$$|\langle x, x_t \rangle| \leq ||x||_2 ||x_t||_2 \leq ||x||_2 \leq \frac{\nu^2}{\gamma}.$$

$\square$

In the following, will use this result to prove any-time confidence estimates for the parameter $\theta$ using the technique of pseudo-maximization, closely following Mutný and Krause (2021).

Recall that $\mathcal{M}_t(x)$ is defined on the ball of radius $\frac{\nu^2}{\gamma L}$. This allows us some freedom in choosing the radius of the ball on which we integrate. In particular, let $K$ be this radius. While we need $K \leq \frac{\nu^2}{\gamma L}$, we can make $K$ larger by choosing larger $\nu^2$ (increasing $\nu^2$ only makes the set of noise distributions larger).

Ultimately, we wish to bound (following Lattimore and Szepesvári (2020))

$$
\begin{aligned}
||\hat{\theta}_t - \theta^*||_{\mathbf{V}_t} &= ||\mathbf{V}_t^{-1} X_{1:t}(X_{1:t}\theta_\star + \eta_{1:t}) - \theta_\star||_{\mathbf{V}_t} \\
&= ||\mathbf{V}_t^{-1} X_{1:t} X_{1:t}\theta_\star - \theta_\star + \mathbf{V}_t^{-1} S_t||_{\mathbf{V}_t} \\
&\leq ||S_t||_{\mathbf{V}_t^{-1}} + \sqrt{\lambda}||\theta_\star||_2.
\end{aligned}
$$

We can not control the second term, so we focus on the first: the self-normalized residuals. Via fenchel duality, one can motivate that the right object to study is the supremum of the martingale $\mathcal{M}_t(x)$ over all $x \in \mathbb{R}$.[5] Define $\tilde{M}_t$ to be the martingale $\mathcal{M}_t$ from above but with $\lambda = 0$, i.e. no regularisation term. Similarly, let $\tilde{\mathbf{V}}_t = \mathbf{V}_t - \lambda\mathbf{I}$ be the design matrix without the regularisation term. Slightly counterintuitively, we will study

$$\bar{M}_t = \int_{||x||_2 \leq K} \tilde{M}_t(x) \mathrm{d}h(x),$$

where $h$ is the probability density function of a truncated normal distribution with inverse variance $\lambda$, that is with covariance matrix $\frac{1}{\lambda}\mathbf{I}$. By Lemma 20.3 in Lattimore and Szepesvári (2020), $\bar{M}_t$ is also a super-martingale with $\bar{M}_0 \leq 1$. Then we have

$$
\begin{aligned}
\bar{M}_t &= \frac{1}{N(h)} \int_{||x||_2 \leq K} \exp\left(x^\top S_t - \frac{1}{2}||x||_{\tilde{\mathbf{V}}_t}^2\right) \exp\left(-\frac{1}{2}x^\top \lambda\mathbf{I}x\right) \mathrm{d}x \\
&= \frac{1}{N(h)} \int_{||x||_2 \leq K} \exp\left(x^\top S_t - \frac{1}{2}||x||_{\mathbf{V}_t}^2\right) \mathrm{d}x.
\end{aligned}
$$

We will define the shorthand $f_t(x) = x^\top S_t - \frac{1}{2}x^\top \mathbf{V}_t x = f_t(x^*) + \nabla f_t(x^*)^\top(x - x^*) - \frac{1}{2}(x - x^*)^\top \mathbf{V}_t(x - x^*)$ (by Taylor's theorem), where $x^* = \arg\max_{||x|| \leq kK} f_t(x)$, $k \in (0, 1)$ will be chosen later. We can lower bound $\bar{M}_t$ by

$$
\begin{aligned}
\bar{M}_t &= \frac{1}{N(h)} \int_{||x||_2 \leq K} \exp\left(x^\top S_t - \frac{1}{2}||x||_{\mathbf{V}_t}^2\right) \mathrm{d}x \\
&= \frac{\exp(f_t(x^*))}{N(h)} \int_{||x||_2 \leq K} \exp\left(\nabla f_t(x^*)^\top(x - x^*) - \frac{1}{2}(x - x^*)^\top \mathbf{V}_t(x - x^*)\right) \mathrm{d}x \\
&= \frac{\exp(f_t(x^*))}{N(h)} \int_{||y + x^*||_2 \leq K} \exp\left(\nabla f_t(x^*)^\top y - \frac{1}{2}y^\top \mathbf{V}_t y\right) \mathrm{d}y \quad\quad (34) \\
&\geq \frac{\exp(f_t(x^*))}{N(h)} \int_{||y||_2 \leq (1-k)K} \exp\left(\nabla f_t(x^*)^\top y - \frac{1}{2}y^\top \mathbf{V}_t y\right) \mathrm{d}y \quad\quad (35)
\end{aligned}
$$

---

[5] But that is in our case ill-defined

$$
= \frac{\exp(f_t(x^*))}{N(h)} \int_{||y||_2 \leq (1-k)K} \exp\left(\nabla f_t(x^*)^\top y\right) \exp\left(-\frac{1}{2} y^\top \mathbf{V}_t y\right) dy
$$

$$
= \frac{\exp(f_t(x^*))N(g)}{N(h)} \mathop{\mathbb{E}}_{y \sim g} \left[\exp\left(\nabla f_t(x^*)^\top y\right)\right]
$$

$$
\geq \frac{\exp(f_t(x^*))N(g)}{N(h)} \exp\left(\mathop{\mathbb{E}}_{y \sim g}[\nabla f_t(x^*)^\top y]\right) \tag{36}
$$

$$
= \frac{\exp(f_t(x^*))N(g)}{N(h)}.
$$

where in step (34) we used the change of variables $x = y + x^*$. In (35) we use that if $||y||_2 \leq (1-k)K$, then $||x^* + y||_2 \leq ||x^*||_2 + ||y||_2 \leq (1-k)K + kK = K$. Finally, in (36), we used Jensen's inequality. The last inequality follows from symmetry. Note that we implicitly defined $g$ to be a truncated normal distribution with covariance matrix $\mathbf{V}_t^{-1}$ on the ball of radius $(1-k)K$.

This puts us in a position to put Ville's inequality to good use:

$$
\delta \geq \mathbb{P}\left(\exists t : \log(\bar{M}_t) \geq \log(1/\delta)\right)
$$

$$
\geq \mathbb{P}\left(\exists t : f_t(x^*) + \log\left(\frac{N(g)}{N(h)}\right) \geq \log(1/\delta)\right)
$$

$$
\geq \mathbb{P}\left(\exists t : f_t(x^*) \geq \log\left(\frac{N(h)}{N(g)\delta}\right)\right).
$$

We now wish to recover $||S_t||_{\mathbf{V}_t}$. Recall the definition of $f_t(x^*)$ as the maximum over all $x$ in a ball of radius $kK$. Consequently, we can choose $x = \frac{\mathbf{V}_t^{-1}S_t}{||S_t||_{\mathbf{V}_t^{-1}}}\sqrt{\lambda}kK$, which has norm bounded by $kK$. We have

$$
f_t(x^*) \geq f_t\left(\frac{\mathbf{V}_t^{-1}S_t}{||S_t||_{\mathbf{V}_t^{-1}}}\sqrt{\lambda}kK\right) = ||S_t||_{\mathbf{V}_t^{-1}}\sqrt{\lambda}kK - \lambda k^2 K^2,
$$

which immediately yields

$$
\mathbb{P}\left(||S_t||_{\mathbf{V}_t^{-1}} \geq \sqrt{\lambda}kK + \frac{1}{\sqrt{\lambda}kK}\log\left(\frac{N(h)}{N(g)\delta}\right)\right) \leq \delta.
$$

The only thing that remains is bounding $\log\left(\frac{N(h)}{N(g)}\right)$.

We give the following Lemma that is a slightly generalized version of Mutný and Krause (2021) and originally inspired by Faury et al. (2020).

**Lemma 14.** *The normalizing constants satisfy*

$$
\log\left(\frac{N(h)}{N(g)}\right) \leq d\log\left(\frac{1}{1-k}\right) + \log\left(\frac{(\det(\mathbf{V}_t))^{1/2}}{\det(\sqrt{\lambda}\mathbf{I})}\right).
$$

We can use the bound from Lemma 14 to conclude that

$$
\mathbb{P}\left(||S_t||_{\mathbf{V}_t^{-1}} \geq \sqrt{\lambda}kK + \frac{d}{\sqrt{\lambda}kK}\log\left(\frac{1}{1-k}\right) + \frac{1}{\sqrt{\lambda}kK}\log\left(\frac{(\det(\mathbf{V}_t))^{1/2}}{\delta\det(\sqrt{\lambda}\mathbf{I})}\right)\right) \leq \delta.
$$

We stated earlier that

$$
||\hat{\theta}_t - \theta^*||_{\mathbf{V}_t} \leq ||S_t||_{\mathbf{V}_t^{-1}} + \sqrt{\lambda}||\theta_\star||_2.
$$

Combining this with our analysis, we get the Proposition 3.

We may now choose the parameters $k$, $K$ and $\lambda$. Note that to get sub-Gaussian rates as in Abbasi-Yadkori, one needs to pick a regularization parameter of the order of $\lambda = d\log(T)$.

**Proof of Lemma 14**   We give a proof of the Lemma for completeness, and because the additional generality makes for a slightly different proof, even though the bound stays the same.

*Proof.* We have

$$N(h) = \int_{||x||_2 \leq K} \exp(-\lambda ||x||_2^2) \mathrm{d}x$$

$$= \frac{1}{|\det(\sqrt{2\lambda}\mathbf{I})|} \int_{||x||_2 \leq K} \exp\left(-\frac{1}{2}||\sqrt{2\lambda}x||_2^2\right) |\det(\sqrt{2\lambda}\mathbf{I})| \mathrm{d}x$$

$$= \frac{1}{|\det(\sqrt{2\lambda}\mathbf{I})|} \int_{||x||_2 \leq \sqrt{2\lambda}K} \exp\left(-\frac{1}{2}||x||_2^2\right) \mathrm{d}x.$$

Further we have

$$N(g) = \int_{||x||_2 \leq (1-k)K} \exp(-\frac{1}{2}x^\top \mathbf{V}_t x) \mathrm{d}x$$

$$= \frac{1}{|\det(\mathbf{V}_t^{1/2})|} \int_{||x||_2 \leq (1-k)K} \exp(-\frac{1}{2}||\mathbf{V}_t^{1/2}x||_2^2) |\det(\mathbf{V}_t^{1/2})| \mathrm{d}x$$

$$= \frac{1}{|\det(\mathbf{V}_t^{1/2})|} \int_S \exp(-\frac{1}{2}||x||_2^2) \mathrm{d}x,$$

where $S = \{\mathbf{V}_t^{1/2}x \,|\, ||x|| \leq (1-k)K\} = \{x \,|\, ||\mathbf{V}_t^{-1/2}x|| \leq (1-k)K\} = \{x \,|\, x^\top \mathbf{V}_t^{-1}x \leq (1-k)K\}$. Note that $\mathbf{V}_t \succeq \lambda \mathbf{I}$ and so $\mathbf{V}_t^{-1} \preceq \frac{1}{\lambda}\mathbf{I}$. Therefore if $||x||_2^2 \leq (1-k)K\sqrt{\lambda}$, we have

$$\sqrt{x^\top \mathbf{V}_t^{-1}x} \leq \frac{1}{\sqrt{\lambda}}||x||_2 \leq (1-k)K \implies x \in S.$$

Thus $\{x \,|\, ||x||_2 \leq (1-k)\sqrt{\lambda}K\} \subseteq S$ and

$$N(g) \geq \frac{1}{|\det(\mathbf{V}_t^{1/2})|} \int_{||x||_2 \leq (1-k)\sqrt{\lambda}K} \exp(-\frac{1}{2}||x||_2^2) \mathrm{d}x.$$

We may therefore bound

$$\frac{N(g)}{N(h)} \leq \frac{(\det \mathbf{V}_t)^{1/2}}{(\det \sqrt{2\lambda}\mathbf{I})} \frac{\int_{||x||_2 \leq \sqrt{2\lambda}K} \exp\left(-\frac{1}{2}||x||_2^2\right) \mathrm{d}x}{\int_{||x||_2 \leq (1-k)\sqrt{\lambda}K} \exp(-\frac{1}{2}||x||_2^2) \mathrm{d}x}.$$

By a rather crude bound (as $1 - k \leq \sqrt{2}$ in any case) we get

$$\frac{\int_{||x||_2 \leq \sqrt{2\lambda}K} \exp\left(-\frac{1}{2}||x||_2^2\right) \mathrm{d}x}{\int_{||x||_2 (1-k)\sqrt{\lambda}K} \exp(-\frac{1}{2}||x||_2^2) \mathrm{d}x}$$

$$\leq \frac{\int_{||x||_2 \leq (1-k)\sqrt{\lambda}K} \exp\left(-\frac{1}{2}||x||_2^2\right) \mathrm{d}x + \int_{(1-k)\sqrt{\lambda}K \leq ||x||_2 \leq \sqrt{2\lambda}K} \exp\left(-\frac{1}{2}||x||_2^2\right) \mathrm{d}x}{\int_{||x||_2 \leq (1-k)\sqrt{\lambda}K} \exp(-\frac{1}{2}||x||_2^2) \mathrm{d}x}$$

$$= 1 + \frac{\int_{(1-k)\sqrt{\lambda}K \leq ||x||_2 \leq \sqrt{2\lambda}K} \exp\left(-\frac{1}{2}||x||_2^2\right) \mathrm{d}x}{\int_{||x||_2 \leq (1-k)\sqrt{\lambda}K} \exp(-\frac{1}{2}||x||_2^2) \mathrm{d}x}$$

$$\leq 1 + \frac{\exp\left(-\frac{1}{2}(1-k)^2 \lambda K^2\right)}{\exp\left(-\frac{1}{2}(1-k)^2 \lambda K^2\right)} \frac{\int_{(1-k)\sqrt{\lambda}K \leq ||x||_2 \leq \sqrt{2\lambda}K} \mathrm{d}x}{\int_{||x||_2 \leq (1-k)\sqrt{\lambda}K} \mathrm{d}x}$$

$$= 1 + \frac{\mathrm{vol}_d(\sqrt{2\lambda}K) - \mathrm{vol}_d((1-k)\sqrt{\lambda}K)}{\mathrm{vol}_d((1-k)\sqrt{\lambda}K)}$$

$$= \frac{\mathrm{vol}_d(\sqrt{2\lambda}K)}{\mathrm{vol}_d((1-k)\sqrt{\lambda}K)}$$

$$= (1-k)^{-d}\sqrt{2}^d.$$

We can put this together to obtain

$$\frac{N(h)}{N(g)} \le (1-k)^{-d}\sqrt{2}^d \frac{(\det(\mathbf{V}_t))^{1/2}}{\det(\sqrt{2\lambda}\mathbf{I})} = (1-k)^{-d}\frac{(\det(\mathbf{V}_t))^{1/2}}{\det(\sqrt{\lambda}\mathbf{I})}.$$

$\square$