# OpenReview forum: "Likelihood Ratio Confidence Sets for Sequential Decision Making"
_NeurIPS.cc/2023/Conference — NeurIPS 2023 poster_

### Official Review · Reviewer_XFVf · 2023-07-06

**Soundness:** 3 good
**Presentation:** 3 good
**Contribution:** 2 fair
**Rating:** 6
**Confidence:** 3

**Summary:**

This paper proposed to use the likelihood ratio approach to provide an any-time valid confidence sequence, which is suitable for problems with well-specified likelihood. It discusses how to provably choose the best sequence of estimators and sheds light on connections to online convex optimization. To counteract the initially large bias of estimators, they propose a reweighing scheme. They also provide a non-asymptotic analysis of the likelihood ratio confidence set size for GLM, and performed numerical simulations.



**Strengths:**

The idea of using the universal inference method in bandit problems is interesting.


**Weaknesses:**

In bandit problems, the target of interest is regret, and the validity/length of the confidence set is a tool of analysis. One shortcoming of this paper is that there is no regret analysis (of the bandit problem) of the proposed method. One important question is, what is the regret of the proposed method on simple bandit problems such as the K-armed bandit (UCB method gives \sqrt{KT} regret) and the linear bandit (LinUCB gives \sqrt{d^2 T} regret). Since bandit is the focus of this paper, without a concrete regret bound (showing that it matches the best existing method), the benefit of the proposed method is not convincing.

The method proposed in this paper is similar to the OMLE method [1], which has a provable regret guarantee in many RL problems. One interesting question is to study whether the proposed algorithm has benefit over the OMLE algorithm.

[1] Optimistic MLE—A Generic Model-based Algorithm for Partially Observable Sequential Decision Making.

After reading the rebuttal:
I am convinced that a regret bound can be obtained for this method. I trust the authors will add this in the revised version.

**Questions:**

NA

---

> ### Author Rebuttal · Authors · 2023-08-09
>
> Thank you for your comments. As you mention, the LinUCB analysis uses the radius of the confidence set to derive the overall regret bound for the bandit problem. Of course, we can do this with our confidence sets as well, and in the linear case, obtain immediately the optimal regret. Indeed, the proof is one line, starting from our confidence set diameter, and multiplying with a $\sqrt{Td\log T}$ factor due to the elliptical potential Lemma (Lemma 9), we obtain a $ \sqrt{Td\log T} \times \texttt{confidence parameter}$ regret bound. Since our confidence sets have size $\mathcal{O}(\sqrt{d\log T})$ we match the lower bounds $\sqrt{T}d$ for the problem up to log-factors for the Gaussian linear model case, i.e. $\sqrt{T}d\log T$. Similar steps can be taken for generalized linear models leveraging smoothness and strong convexity constants. We can add these theoretical results in a revised version.
>
> As to the OMLE paper, this work also uses likelihood ratios to define the confidence set for a family of likelihoods. **However the method significantly differs from our setup.** *They do not use the online prediction game as a comparator for the likelihood ratio in the denominator.* Instead, they use a running MLE estimator as a comparator – not a sequence of estimators viewed as a game. This way they have to resort to the same type of analysis as prior works, say for sub-Gaussian likelihoods when calculating the confidence parameter $\beta$, and need to adopt the worst-case perspective.
>
> In our work, we do not depend on these worst-case analyses, since the confidence set radius is defined *implicitly*, by way of the real-time performance of the online sub-learner, and not its worst-case performance guarantee. This is a **crucial distinction** between our work and other approaches, including OMLE and online-to-confidence set conversions. Additionally, our method can be applied to any likelihood without the need to perform a worst-case/theoretical analysis to get the right confidence parameter *at all*.
>
> We invite you to reconsider your score, as our work brings significant theoretical and practical contributions on top of the paper you mention, and we are convinced that the similarities are only superficial.

---

> > ### Comment · Reviewer_XFVf · 2023-08-15
> >
> > Thanks for the clarification of the authors. I am satisfied that the regret bound can be obtained for the linear bandit, and I trust that the authors will add it to the revised version. I will raise my score.

---

### Official Review · Reviewer_4g2j · 2023-07-06

**Soundness:** 3 good
**Presentation:** 1 poor
**Contribution:** 3 good
**Rating:** 6
**Confidence:** 4

**Summary:**

This paper examines the confidence set of an estimator, defined by a likelihood ratio. The contributions stated within the work, alongside my corresponding queries, are outlined as follows:
* For generalized linear models, we theoretically analyze the geometry of the LR confidence sets under mild assumptions. We show their geometry is dictated by Bregman divergences of exponential families (Chowdhury et al., 2022).
*We show that the size of the confidence set is dictated by an online prediction game. The size of these sets depends on a sequence of estimators \{\theta_s\}_{s=1}^t that one uses to estimate the unknown parameter \theta_*. We discuss how to pick the estimator sequence in order to yield a provably small radius of the sets, by using the Follow-the-Regularized-Leader algorithm, which implements a regularized maximum-likelihood estimator. We prove that the radius of the confidence sets is nearly-worst-case optimal, and accordingly they yield nearly-worst-case regret bounds when used in generalized linear bandit applications. However, due to their data-dependent nature, they can be much tighter than this theory suggests.
*We analyze limitations of classical (un-weighted) LR sets when the underlying conditional observation model is not identifiable. In this case, the resulting (inevitable) estimation bias unnecessarily increases the size of the confidence sets. To mitigate this, we propose an adaptive reweighting scheme that decreases the effect of uninformed early bias of the estimator sequence on the size of the sets downstream. The reweighting does not affect the coverage guarantees of our sets, and utilizes an elegant connection to (robust) powered likelihoods (Wasserman et al., 2020).
* Thanks to the adaptive reweighting scheme, our sets are very practical as we showcase experimentally. We demonstrate that our method works well with exponential and non-exponential family likelihoods, and in parametric as well as in kernelized settings. We attribute their practical benefits to the fact that they do not depend on (possibly loose) worst-case parameters.

**Strengths:**

The research question posed in this paper is notably intriguing and holds significant potential for various applications. Furthermore, the innovative approach adopted by the authors represents a substantial contribution to the field.

**Weaknesses:**

While the paper offers several valuable insights, I noticed that the overall integration of its contributions could be improved. For instance, claimed contributions 1 and 3, which could potentially have been effectively merged, were instead treated separately. This compromise the contribution significantly from my perspective. Furthermore, I have concerns about the clarity and accuracy of some claims. For more specific observations, please refer to the "Questions" section. I believe addressing these issues would greatly enhance the coherence and validity of the work.

**Questions:**

* The relationship between LR confidence set geometry and Bregman divergences is briefly discussed on page 6. It would be beneficial to have this connection elaborated upon more thoroughly in the main body of the text.

* Section 2 provides a clear explanation on how to select a sequence of $\{ \theta_s \}_{s=1}^t$ to minimize the radius of the confidence set. However, this connection becomes less clear in Section 3.

Moreover, the paper claims that the radius of the confidence set is nearly-worst-case optimal. Could you please provide further explanation as to why this is the case? While it's clear that the FTRL can lead to nearly-worst-case optimal regret for the online optimization problem, $\mathcal{R}_t$ defined in $(3)$ appears to be not identical, but smaller than the definition of regret in the online optimization problem. If $\mathcal{R}_t$ is significantly smaller than the regret of the FTRL algorithm, might our confidence set not be optimal?

* The introduction of a re-weighting scheme is intriguing. However, given that the re-weight $w_t$ is difficult to compute, could you provide any theoretical performance guarantees? Additionally, in Theorem 2, the bias estimate depends on \norm{\theta_*}_2^2. Could this potentially conflict with our objective to estimate the confidence set of \theta_*? Furthermore, in Section 3, we ascertain the geometry of the linear models assuming $w_t = 1$. Is it possible to perform a similar analysis incorporating the re-weighting scheme? Otherwise, the value of the re-weighting scheme might be discounted if we cannot incorporate it into the analysis of simple model

* A query arises from Theorem 3, where \theta \in \mathcal{C}_t satisfies a certain inequality. I am curious about the application of this inequality in estimating the confidence set of \theta_*, given that the Bregman distance involved depends on $\theta_*$, which is unknown. Is this result primarily of theoretical interest, or is it of practical use in this context?

---

> ### Author Rebuttal · Authors · 2023-08-09
>
> Thank you for your feedback on improving the clarity and coherence of our paper. We very much welcome the fact you see our contribution as substantial and we will use your insights to improve the exposition. The reason we separated the exposition of our first and third contribution is because we see the theory and methodology as separate contributions. The theory for GLMs and the proposition of a reweighting scheme can be considered adjacently, as reweighting is a more general approach that is not limited to GLMs.
>
>  We now answer your individual questions in order:
>
> **Q:** *Bregman divergence and confidence sets* **R:** The relationship between Bregman divergence and confidence set is indeed intriguing and we are planning to add more information in a revised version. The main reason it arises is in the study of exponential family likelihoods that generalize the Gaussian case. In the Gaussian case, we can measure the confidence set in an ellipsoid norm, and in GLMs, Bregman divergences are the appropriate generalization. We plan to add more 2d confidence sets for different exponential families to provide intuition in the revised version.
>
> **Q:** Selecting online learners **R:** Any low regret online learning algorithm can be chosen to select $\\{\hat{\theta}_t\\}_t$ as you can see in Eq. (9). We make the FTRL choice because of its well-specified meaning as a regularized maximum likelihood estimator, but considering algorithms based on exponential weights or other types of forecasters is possible too.
>
> **Q:** *Regret notions* **R:** The two notions of regret are indeed different, but lower bounds on linear regression show that the difference between the two can be at most logarithmic. Furthermore, a good way to convince ourselves that our confidence sets are nearly optimal at least in the Gaussian case is to look at the downstream linear bandit regret. It follows easily from our confidence set size and specification that we achieve optimal regret in a linear bandit application. Therefore, the set being any smaller would imply that we broke through a lower bound for linear bandits, an impossibility.
>
> Additionally, we would like to point out an *important distinction* between our work and other types of online-to-confidence set conversions. Here, we do not need a regret bound for the online learner to implement our algorithm. Our confidence set will simply *adapt* to the performance of the learner we use. It is therefore not crucial to understand the sharpest possible rates of FTRL on this slightly easier problem. Our theory is mostly derived for validation, but is not necessary for implementation.
>
> **Q:** *The introduction of a re-weighting scheme is intriguing.* **R:** The reweighting we propose can be difficult to compute in general, but does not have to be. For example, for GLMs, or any likelihood satisfying Ass of Thm. 2 the calculation is rather straightforward. The reweighting scheme is a very important addition for practical purposes, see the gray arrow in Figure 2a). This scheme is a crucial ingredient in making our method competitive with heuristics.
>
> **Q:** *“In Theorem 2, the bias estimate depends on* $||\theta_*||$ *conflict with our objective to estimate the confidence set of* $\theta_*$?*”* **R:** In Theorem 2, the bias does indeed depend on the value of $\theta_*$, but for our purposes an upper bound on its norm is sufficient. Namely, we merely require having $||\theta_*||_2^2 \leq B^2$ throughout the paper (Assumption 1), which is necessary knowledge for competing methods as well. Making the bound loose for the purposes of balancing bias just hampers the sharpness of the confidence sets and not the coverage guarantees.
>
>
> **Q:** *Incorporating the reweighing* **R:** Incorporating the reweighing scheme in Section 3 is a challenging problem. We were unable to provide do so, and we find this extension challenging. While the geometry of the set is influenced by the weighting (effects we do not analyze), the size of the set is analyzed with the weighting scheme as well, which gives us the intuition. Arguably, the weighting scheme is in place to counteract undesirable effects of bias, which inflates the regret of the online learner. Therefore, it arguably makes sense to analyze the effects of weighting on the online learning regret. We do agree that in principle though, it would be very interesting to understand the geometry at an even more precise level. For purely theoretical purposes, the weighting can be discarded, since up to constant factors, the resulting set will be equivalent with and without it. However, reweighting observations is of tremendous practical importance, in particular in non-parametric settings, so we are convinced of the scheme's value nonetheless. Thanks to the weighting scheme, our sets should be the go-to practical method in cases where the likelihood is well-specified.
>
> First and foremost, we wish to emphasize that Theorem 3 is not needed for the construction of the sets in any way. The theorem is merely there for theoretical understanding. The reason the Bregman divergence depends on $\theta_*$ is that the likelihood landscape (geometry) is influenced by the true data-generating process beyond the Gaussian case. In case one desires global bounds, a worst-case perspective could be taken.

---

> > ### Author Response · Authors · 2023-08-17
> > **Discussion period ends soon**
> >
> > Dear reviewer, the discussion window is closing soon. We believe we have addressed your questions and concerns and are happy to engage in further discussion. Please reconsider your score.

---

> > ### Comment · Reviewer_4g2j · 2023-08-20
> >
> > I concur with the authors regarding their well-articulated responses, which addressed my queries. As a result, I am inclined to raise my review score. Nonetheless, I share the sentiment expressed by some of my fellow reviewers that the manuscript would benefit greatly from a thorough revision for readability. Enhanced clarity and structure could not only facilitate better comprehension but also underscore the significance of the paper's contributions.

---

### Official Review · Reviewer_DGax · 2023-07-07

**Soundness:** 3 good
**Presentation:** 3 good
**Contribution:** 3 good
**Rating:** 6
**Confidence:** 3

**Summary:**

The paper proposes to use likelihood ratios to construct confidence sequences that facilitate the downstream online decision making under uncertainty. The weighting and corresponding bias estimation are proposed to avoid regret blow-up in low-noise setting. The paper offers theoretical insights for the bias estimation, the geometry of the confidence set for generalized linear model (GLM), and regret in the online optimization in GLM. The paper offers the corresponding empirical results on bandits problems.

**Strengths:**

1. The paper incorporates the likelihood ratios into online decision-making. The LR confidence set reduces the requirement of conventional concentration results and allows existing algorithms to solve more generalized problems as long as the likelihood is well specified. There is room for studying its application in various sequential decision-making tasks.
2. The adaptive weighting scheme is well justified with practical bias estimation.
3. The geometry analysis could be helpful to justify the usage of the LR confidence set rather than alternatives.

**Weaknesses:**

1. In general the paper is clearly organized. However, the scattered yet closely related notations in sections 2 and 3 make it challenging to follow. I would recommend further highlighting important definitions and formulas or trying to concentrate on them to allow easier revisits.
2. The experiment results shown in Figure 2 lack statistical significance. The fact that the figure shows median values rather than mean values indicates the stability problem of the experiment results.
3. The mismatch between the theoretical results in section 3 with regards to the online optimization and the empirical study mostly focused on bandit problems in section 4. As only the downstream cumulative regrets are shown.

**Questions:**

1. Minor issue: the product on the denominator of eq (1) seems to be falsely indexed, could the author double-check?
2. In the bandit problems, how are the confidence levels specified?
3. Could the author briefly comment on the potential impact of misspecified likelihood (robustness to misspecification when there is a uniform error bound for the misspecification)?

**Limitations:**

Discussed in the comments above.

---

> ### Author Rebuttal · Authors · 2023-08-09
>
> Thank you for your comments, we will try to polish notation and improve the readability of the manuscript. Now to specific concerns and questions:
>
> **C:**: *"The experiment results shown in Figure 2 lack statistical significance."* **R:** We provide the same plot with the mean in the attached document. However, we disagree with you that plotting the median shows instability. What shows instability is the large spread in general. Plotting quantiles with mean is to the best of our knowledge non-standard, hence the median was chosen. We find results reporting mean more questionable. Additionally, we would like to point out that our method is stable and instability usually arises with heuristics such as MK2021 or NR2020 which suffer from numerical issues.
>
> **C:** *“Mismatch”* **R:** At first sight, there might seem to be a mismatch between theory and experiments, however, downstream bandit regret and the size of the confidence set are intimately related. For the UCB algorithm (the one we use in the experiments), the better the confidence sets, the better the algorithm's bound and its practical performance. We chose to demonstrate our applicability in this manner, as at the end of the day, our sets' worth is measured by their downstream applicability. This being said, we prove a calibration plot in the attached documents and will add it to the paper. The plot shows that the heuristic methods lose coverage while the other provable methods are way too conservative. These likelihood ratios are somewhere between these two extremes. The calibration plots come with caveats however, since they depend on the true parameter $\theta_*$, and how you collect the data. Since we are not interested in the iid setting, we need to choose a way to collect data, say via a bandit algorithm or in some active learning setting. In these scenarios, calibration plots are rarely what people are interested in, and therefore a rather non-standard method of evaluation. We show our results for two different values of $\theta_*$.
>
> We understand that the different regret notions (bandit application regret vs. regret of the online learner) could lead to confusion, and we will clarify this in a revised version. In terms of theory, for that reason as well, we were reluctant to state explicit regret bounds of the bandit algorithm. Note however, that they follow immediately by an application of the elliptical potential lemma. For instance, for linear bandits, we directly obtain the optimal $\tilde{\mathcal{O}}(d\sqrt{T})$ regret rate.
>
> **Q1:**: Thank you for pointing this out. This is indeed a typo, and the index should be $s$ instead of $i$.
>
> **Q2:** The confidence levels are done exactly as proposed in the paper and maintain provable coverage up to the chosen $\delta$. We chose $\delta = 0.1$. We compare the exact methods from the literature as well as to empirical heuristics popular in the field that people use in deployed applications. The reason our sets need no tuning is that the confidence radius is implicit when compared to ellipsoid confidence sets from the literature with fixed (worst-case dependent) radius. Since we practically use no inequalities to derive our confidence sets, they are arguably closer to what is practically achievable with provable coverage, and inherit forms of instance-dependency.
>
> **Q3:** We do not discuss robustness in the paper, and a proper treatment requires a follow-up. In general, as noted in the work on universal inference [1], exponentiating a likelihood with small powers can improve robustness, but this is only a heuristic. In general, in the presence of misspecification, it is not possible to guarantee coverage with our approach. In contrast, the size analysis mostly depends on tail properties of the random variables involved, and it is believable that this part can be wholly extended accordingly.
>
> We spent some time thinking about probably robust versions of our approach, but did not succeed. In case robust variants are necessary, confidence sets which are designed for sub-families may be more appropriate e.g. [1].
>
> *[1] Yadkori, Pal,Szepesvari (2012) Improved algorithms for linear stochastic bandits*

---

> > ### Comment · Reviewer_DGax · 2023-08-15
> >
> > I appreciate the authors' detailed response. I believe my questions have been well answered, and incorporating the corresponding discussion into the paper would enhance the overall presentation. I have carefully considered the responses to my concerns about the mismatch between the theoretical results and the later empirical study on regrets of bandit problems, as similar questions have also been raised by other reviewers. While I believe the answers provided could alleviate the issue, I still hold the belief that establishing a more direct connection to justify the benefits of the proposed method in the bandit problem is crucial for the paper's coherence. In general, I value the theoretical contribution of the paper, but I do have concerns about its completeness and presentation, which could potentially be better addressed in the revised version. Therefore, I will maintain my current score.

---

> > > ### Author Response · Authors · 2023-08-16
> > >
> > > Thank you for engaging in the discussion. We would like to respectfully disagree with your conclusions:
> > >
> > > - We provided the arguably most useful demonstration of improvement in bandit problems by showing that empirically the regret decreases when using our method. This is almost always the relevant and targeted metric in the field.
> > > - As we discussed, the mismatch the reviewer is pointing out is inherent for adaptive sequences. One first needs an algorithm to generate the adaptive sequence which makes the generation of calibration plots application dependent. We believe they are not the optimal way to showcase our method accordingly, which is why we refrained from doing so in the first place.
> > > - We provided calibration plots in response to the review nonetheless. One last possible addition would be to directly plot the radius of the confidence sets and compare them to a baseline ellipsoidal set. However, this is only satisfactory in the Gaussian case as it‘s the only case where our sets are actually ellipses -- essentially our banner plot. We are interested to know if the reviewer would be satisfied with this.
> > > - We provided an explanation that a regret bound follows automatically from our confidence sets. However, the main contribution is not improving bandit rates. We do not improve any known rates. We match the existing rates via our universal setup.
> > >
> > > We invite the reviewer to look at this paper more holistically instead of looking purely for theoretical regret bounds. This paper provides a novel and unique way of looking at the problem of adaptive confidence sets in machine learning applications. We hope that matching regret rates in known settings will encourage practitioners to use our methods in different and novel settings and enjoy the empirical benefit of this provable method.

---

### Official Review · Reviewer_CfDQ · 2023-07-07

**Soundness:** 4 excellent
**Presentation:** 4 excellent
**Contribution:** 4 excellent
**Rating:** 8
**Confidence:** 3

**Summary:**

This paper proposes a new construction of confidence sets for a parametric setting, where the likelihood of the noise process is explicitly given.
The proposed method is based on a weighted variant of the sequential likelihood ratio (LR) statistics, which was proposed in universal inference (Wasserman et al., 2020).
The confidence set is a function of the choice of estimators that construct the sequential LR statistics, and the paper proposes a specific method to construct such estimators by the means of Follow-the-Regularized-Leader (FTRL) algorithm in online learning, by viewing the confidence parameter minimization as a regret minimization problem.
Further, the paper proposes a new adaptive reweighting scheme, which uses a powered LR statistics to make the resulting confidence set more “robust”. The proposed reweighting scheme is based on the bias of the used estimators. While the bias may not be directly estimated, they provide a computable upper bound on the bias for a certain form of distributions.
For generalized linear models, the size of the confidence set is analyzed under regularity assumptions. To make the bound fully concrete, they also provide a high-probability regret guarantee of the used FTRL estimators (for GLMs).
The effectiveness of the proposed method is demonstrated with linear and kernelized bandit experiments.


**Strengths:**

The paper is very well written. The proposed framework and analysis are a nice and careful combination of several different techniques (universal inference, FTRL, Bregman information gain, …) in statistics and online learning.
I think this could be indeed useful in bandit problems but it can be also applicable whenever parametric models are available.


**Weaknesses:**

I do not find a crucial weakness of this paper, except a few questions detailed below.


**Questions:**

- I find the sentence in line 107 a bit confusing. It says “If the confidence parameter goes to zero, only $\theta_\star$ is in the set.” This is not the case, as the LHS (of the inequality inside the set notation) in the equation after line 103 is 1 if $\theta=\theta_\star$, and so $\theta_\star$ will be excluded if the RHS goes to zero. What should be a proper explanation?
- I cannot make much sense of the specific reweighting scheme in line 162 right away, though I can understand the intuition thanks to the Gaussian example. Can you elaborate this choice?

A possible typo:
- Line 153: depending only $\theta_\star$



**Limitations:**

Limitations are not discussed.

---

> ### Author Rebuttal · Authors · 2023-08-09
>
> Thank you for your comments. Indeed you are right; this is a typo on our side. What we meant to say is if the $\log$ of the expression goes to zero (or the expression goes to 1), then only the true parameter $\theta^*$ is included in the set. We bear in mind here that the prediction game is played on the log-likelihood loss.
>
> Regarding the reweighting: In general, there are two sources of error, stochastic (sometimes called "aleatoric" uncertainty) and deterministic (sometimes called "epistemic" uncertainty) – we refer to it as bias. The likelihood function properly measures only stochastic noise. In a certain sense, our problem is not well posed in the absence of sufficiently rich data. For example, it is impossible to distinguish between (linear) models that differ only in the subspace orthogonal to the data. This causes the likelihood ratio to unnecessarily blow up, as we show in theory and in examples.
>
> Let us now give some intuition behind the choice of our reweighting scheme. A naive solution would be to gather data and set $w_i = 0$ until the bias vanishes -- i.e., we observe data that span the whole space. However, this might never happen in non-parametric settings like RKHSs, and is therefore not viable. Additionally, even when the bias is non-zero, there is information that can be leveraged by including it in the likelihood ratio sequence. Now, the question is how much information, and how can we include it. The reasoning we adopt is motivated by understanding how much the deterministic error (bias) influences the likelihood ratio when compared against the stochastic component. If these two parts are, say, equal in magnitude, then we would choose $w_i =1/2$ to reflect that only half of the contribution is the stochastic part relevant to the likelihood formulation. This simple example, generalizes the intuition from the Gaussian case to GLMs, and helps us to reduce the bias of the online optimizer that we analyze in Sec. 3.2.
>
> Given that we are interested in practical methods, and not just theory, we were driven to propose a widely applicable and performant scheme. Our reweighting scheme is just that: motivated by balancing out terms in the regret of the online learner (by effectively changing its objective), it strikes a balance between bias and variance.

---

> > ### Comment · Reviewer_CfDQ · 2023-08-16
> >
> > I appreciate the authors' response and I read other reviews as well. I think that the authors can incorporate the changes in the revision to make the contributions clearer. Especially the regret bound for linear bandits (as pointed out by Reviewer XFVf) and a better intuitive explanation on the reweighting scheme will strengthen the paper. Hence I will keep the score.

---

### Author Rebuttal · Authors · 2023-08-09

We would like to thank all reviewers for their time and effort spent. We are pleased that most reviewers see the benefits of our work. We hope that our individual responses clarify any misunderstandings and that the reviewers will consider raising their scores if they see fit.

We really believe this method is of great importance to the fields utilizing adaptive inference, and hope that we will get the opportunity to present it to the community. We attach some additional plots for reviewer DGax.

---

### Decision · Program_Chairs · 2023-09-21

**Decision:**

Accept (poster)

**Comment:**

This paper uses a likelihood ratio approach to construct any-time valid confidence sequences for downstream sequential decision making applications. The approach is based on a weighted variant of sequential likelihood ratio (LR) statistics (Wasserman et al., 2020), and builds a confidence set based estimators that construct sequential LR statistics. The paper proposes a specific method to construct such estimators using the Follow-the-Regularized-Leader (FTRL) algorithm in online learning, viewing the confidence parameter minimization as a regret minimization problem, and provide non-asymptotic guarantees, as well as experiments in which the confidence sets are applied to linear and kernelized bandits.

Reviewers generally agreed that the paper's approach is novel and potentially significant. In particular, the results involve an interesting and non-trivial combination of several different techniques (universal inference, FTRL, Bregman information gain, …) from statistics and online learning, and hence may be of interest to multiple communities. As such, there was a consensus that this work should be accepted.

For the final version of the paper, the authors are encouraged to incorporate the reviewers' suggestions, particularly regarding provable regret bounds for the bandit application.